# Transcriptomics Reveal the Effects of Breeding Temperature on Growth and Metabolism in the Early Developmental Stage of *Platax teira*

**DOI:** 10.3390/biology12091161

**Published:** 2023-08-23

**Authors:** Ming-Jian Liu, Jie Gao, Hua-Yang Guo, Ke-Cheng Zhu, Bao-Suo Liu, Nan Zhang, Jin-Hui Sun, Dian-Chang Zhang

**Affiliations:** 1College of Fisheries, Tianjin Agricultural University, Tianjin 300384, China; 2South China Sea Fisheries Research Institute, Chinese Academy of Fishery Sciences, Key Laboratory of South China Sea Fishery Resources Exploitation and Utilization, Ministry of Agriculture and Rural Affairs, Guangzhou 510300, China; 3Southern Marine Science and Engineering Guangdong Laboratory, Guangzhou 511458, China; 4Sanya Tropical Fisheries Research Institute, Sanya 572018, China; 5Guangdong Provincial Engineer Technology Research Center of Marine Biological Seed Industry, Guangzhou 510300, China

**Keywords:** *Platax teira*, temperature, high-throughput sequencing, metabolism, TCA cycle, PPAR pathway

## Abstract

**Simple Summary:**

Temperature plays a crucial role in the growth and development of fish, particularly during the early stages. This study aimed to investigate the effects of different breeding temperatures (21, 24, 27, and 30 °C) on the growth and nutrient metabolism of *Platax teira* larvae using transcriptomic techniques. The analysis revealed differentially expressed genes (DEGs) associated with various biological processes, such as metabolism, catalytic activity, and cellular processes. Pathway analysis showed enrichment in metabolic pathways, including matter and energy metabolism, protein digestion and absorption, and glucose and lipid metabolism. The study also observed that gene expression related to energy, lipid, and glucose metabolism was upregulated at lower temperatures (21 °C), while extremely high temperatures (30 °C) led to suppressed nutrient metabolism and growth due to oxidative stress. These findings highlight the involvement of nutrient metabolism pathways in the thermal stress response of *P. teira*, with an optimal breeding temperature range of 24–27 °C. This study contributes to our understanding of the regulatory mechanisms of larval development under different growth temperatures, providing insights for industrial breeding practices.

**Abstract:**

The growth, development, and survival of fish, especially in the early stages of development, is influenced by a complex of environmental factors, among which temperature is one of the most important. Although the physiological effects of environmental stress in fish have been extensively studied, the molecular mechanisms are poorly understood. However, recent advances in transcriptomic techniques have facilitated the study of the molecular mechanisms of environmental stress responses in aquatic species. Here, we aimed to elucidate the effects of breeding temperatures (21, 24, 27, and 30 °C) on the growth and nutrient metabolism in the early developmental stage of *Platax teira*, using transcriptomic techniques. Transcriptomic analysis identified 5492, 6937, and 4246 differentially expressed genes (DEGs) in the 21 vs. 24 °C, 27 vs. 24 °C, and 30 vs. 24 °C comparisons, respectively, most of which were involved in cell processes, single organism, metabolism, catalytic activity, and cell part, based on gene ontology (GO) functional annotations. Kyoto Encyclopedia of Genes and Genomes (KEGG) enrichment analysis showed that the DEGs were mainly enriched in pathways related to metabolism of matter and energy, protein digestion and absorption, and glucose and lipid metabolism. Additionally, the expression of genes related to energy, lipid, and glucose metabolism in the fish liver was upregulated under a low-temperature condition (21 °C), although increasing the temperature within the acceptable threshold improved nutrient metabolism and growth in the fish. Meanwhile, nutrient metabolism and growth were suppressed by an extremely high temperature (30 °C) owing to oxidative stress. Overall, it was shown that nutrient metabolism pathways were involved in thermal stress responses in *P. teira*, and the optimal breeding temperature range was 24–27 °C. Through transcriptomics, the regulatory mechanism of larval development in *P. teira* under different growth temperatures was elucidated, with the goal of establishing a theoretical basis for industrial breeding.

## 1. Introduction

In recent years, coastal and intertidal zones have experienced an increase in extreme weather events associated with global warming. These events encompass various phenomena, such as severe storms, heatwaves, and abrupt temperature fluctuations [1]. The aquaculture industry, which depends on stable environmental conditions, is particularly susceptible to these changes. Temperature fluctuations have the potential to disturb the growth, reproduction, and overall health of cultured species, resulting in significant economic losses [2]. Consequently, the aquaculture industry faces significant challenges in adapting to and mitigating the impacts of these increasing extreme weather events as-sociated with global warming. Fish are ectothermic animals, and temperature plays a crucial role in maintaining the stability of their internal environment [3]. Fish can respond to temperature changes through physiological, biochemical, and metabolic adaptation processes [4]. Temperature changes above the biological threshold will not only increase the oxygen demand, metabolic rate, and oxygen consumption rate of the organism, but also reduce food intake and inhibit immunity, which can negatively affect growth and survival [5]. In particular, the larvae and juvenile stage are crucial periods, as the organisms are more susceptible to changes in environmental temperature during these stages [6]. Thermal stress is mainly manifested as high deformity rates, low survival rates, and abnormal growth rates of juveniles [7]. Although several studies have examined the physiological response of aquatic species to temperature, the underlying molecular mechanisms are poorly understood.

Recent advances in sequencing and omics technology have facilitated the study of biological mechanisms. High-throughput RNA-seq technology enables the comprehensive analysis of differentially expressed genes (DEGs) under different treatments, improving the understanding of the entire transcriptome of organisms [8]. RNA-seq has been widely used in different fields of aquatic biology. For instance, Qian et al. [9] identified the genes and pathways involved in thermal stress responses in large yellow croaker (*Larimichthy crocea*) using RNA-seq. Similarly, Lyu et al. [10] sequenced the transcriptome of Korean rockfish (*Sebastes schlegelii*) under acute cold and heat stress and identified a total of 584 annotated DEGs. The liver is one of the most important metabolic organs in fish, and it plays an essential role in regulating immune defense and hormone synthesis in fish [11]. Additionally, the liver plays an important regulatory role in thermal stress management by reducing thermal stress damage in fish [12]. Transcriptomic analysis has shown that DEGs in the liver of Nile tilapia (*Oreochromis niloticus*) exposed to cold stress were mainly enriched in nucleic acid synthesis pathway, amino acid and carbohydrate metabolism, apoptosis, and immunity [13]. Similarly, amino acid, fat, and glucose metabolism increased in the liver of spotted perch exposed to high temperatures to meet the energy demand to cope with heat stress [14].

*Platax teira*, also known as long-finned batfish, is native to the Indo-Pacific and is widely distributed in subtropical and tropical coastal areas. *P*. *teira* has a high growth rate and high economic value [15]. However, the growth and development of the species is affected by temperature extremes, especially during the early growth stages. We speculate that the growth, development and survival of the larva and fry of *P. teira* would be affected by environmental temperature. Therefore, this study aims to elucidate the effect of breeding temperature on early development and nutrient metabolism in *P. teira*, using transcriptomic techniques. It is anticipated that the findings of this study would improve the understanding of the molecular mechanism of thermal stress responses in the larva and juveniles of *P. teira*. The results of this study have important implications for the selection of optimal temperature conditions for *P. teira* culture in aquaculture.

## 2. Materials and Methods

### 2.1. Experimental Materials and Design

The *P. teira* (3 years old) used in this study were selected from the brood stock cultivated in Lingshui, Hainan Province. After 45 days of nutrient fortification (during the intensification process, fresh frozen small miscellaneous fish were fed four times a day), 10 female and 10 male fish with mature gonads were selected and anesthetized using eugenol (30–50 mg·L^−1^), and then injected with oxytocin to induce egg release. Thereafter, the eggs were artificially fertilized and transferred into a well-aerated 500 L conical-bottom hatching bucket containing saline water. Floating high-quality fertilized eggs (appearance of myomere) were screened and transport to Shenzhen by plane.

The experiment was performed in four farming ponds (8 m × 7 m × 1.8 m) located in the factory workshop in Shenzhen. The temperature of each culture pond was controlled by a thermostat (Foshan Wino refrigeration equipment Co., Ltd. Foshan, China). Approximately 10,000 fertilized eggs were selected for incubation in the hatching bucket. Based on the hatching results, the hatching rate was calculated to be 95%. The incubation conditions were temperature, 24 °C; dissolved oxygen, >6.5 mg·L^−1^; salinity, 34; and pH, 7.8 ± 0.5. The larvae were cultured in still water for the first 7 days, and then cultured in micro-flow water. Through preliminary experiments, it was found that the larvae of the *P. teira* could not survive at a temperature of <20 °C. To examine the effect of thermal stress on larva and juvenile development, we established a low-temperature (21 °C) and two high-temperature groups (27 and 30 °C), with the natural water temperature (24 °C) as the control group. Each breeding pond contained three hanging cages (1 m × 1 m × 1.5 m) for replication. The hatched larvae were transferred to cages, with 500 fish per cage. The water temperature was adjusted to the experimental temperature at a rate of 1 °C per 8 h from 24 °C, and the water temperature was controlled within ±0.5 °C. Except for the temperature, the cultivation and incubation conditions were the same across the groups, and the fish were fed three times a day, ensuring sufficient food and satiation feeding. The feeding strategy is shown in Figure 1. Every day before feeding, 10 fish were selected from each temperature group to observe the changes in morphology and development. The newly hatched larvae are transparent in appearance, with a considerable distribution of branchiostegal rays on the body surface. They sequentially go through the fry stage and the juvenile stage before entering the young fish stage. During the young fish stage, the fin spines and fin rays have fully developed, and the characteristic features include three black stripes on the head, trunk, and tail, while the body appears silver-white. After 32 days (all temperature groups entered the juvenile stage), a total of 9 fish were taken from each temperature group, 3 fish were randomly selected from each cage and anesthetized with eugenol, and 3 fish liver samples were pooled as one sample, frozen in liquid nitrogen, and stored in a −80 °C refrigerator [16]. All experiments in this study were approved by the Animal Care and Use Committee of the South China Sea Fisheries Research Institute, Chinese Academy of Fishery Sciences (no. SCSFRI96-253) and performed according to the regulations and guidelines established by this committee.

### 2.2. RNA Extraction and Sequencing

Total RNA was extracted from the liver samples using TRIzol reagent (Invitrogen, Thermo Fisher Scientific, Waltham, MA, USA), according to the manufacturer’s instructions. Total RNA concentration and integrity were determined using a NanoDrop (Thermo Fisher Scientific). In this study, when the RNA Integrity Number (RIN) value is greater than 7, the analyzed RNA samples were considered to have sufficient integrity and stability. Input RNA samples were prepared using 3 µg of RNA. mRNA was purified from total RNA using poly-T oligomagnetic beads, and DNA fragmentation was performed at high temperatures using Illumina proprietary fragmentation buffer. First-strand cDNA was synthesized using random oligonucleotide primers and SuperScript II, followed by the synthesis of second-strand cDNA using RNase H and DNA polymerase I. An exonuclease/polymerase activity is required to convert the remaining overhangs to blunt ends and to eliminate the enzyme. Hybridization was performed with Illumina PE oligonucleotides after they were adenylated at the 3′ ends of the DNA fragments. A 400–500 bp cDNA fragment was selected, and the library fragment was purified using the AMPure XP system (Beckman Coulter, Pasadena, CA, USA). DNA fragments were selectively enriched for both end-joined molecules in a 15-cycle PCR using an Illumina PCR primer cocktail. PCR products were purified (AMPure XP system) and quantified using an Agilent Bioanalyzer 2100 (Agilent Technologies, Santa Clara, CA, USA). Libraries were sequenced by Meiji Biotechnology Co., Ltd. using a NovaSeq 6000 platform (Illumina. Inc., San Diego, CA, USA).

### 2.3. Differential Gene Expression Analysis

Trinity (https://github.com/trinityrnaseq/trinityrnaseq/wiki (accessed on 23 January 2023)) was used to assemble the transcriptome from scratch. TransRate (http://hibberdlab.com/transrate/ (accessed on 23 January 2023)) was used to filter and optimize the sequence of the transcriptome and remove common errors (including chimeras, structural errors, incomplete assembly, base errors, etc.). BUSCO (Benchmarking Universal Single-Copy Orthologs, http://busco.ezlab.org accessed on 23 January 2023) was used to evaluate the assembly integrity of the transcriptome. All transcripts obtained by transcriptome sequencing were compared with six major databases (NR, Swiss-Prot, Pfam, COG, GO, and KEGG databases) to obtain annotation information in each database, to help identify and annotate novel genes and transcript isoforms [17]. Differential analysis was performed using DESeq (1.30.0). Genes with an absolute log2 fold change > 1 and a significant *p*-value < 0.05 were classified as differentially expressed genes (DEGs) [18]. If a *p*-value adjustment was applied, we used the adjusted *p*-value instead of the unadjusted *p*-value, along with specifying the type of correction. *p*-values were corrected using the Benjamini–Hochberg procedure.

Two-way clustering analysis was conducted using the Pheatmap (1.0.8) package in the R language [19]. The distances between expression levels of the same gene in different samples and the expression patterns of different genes in the same sample were calculated. The Euclidean and complete chain methods were employed for clustering the data, resulting in the generation of heatmaps.

### 2.4. Bioinformatics Analysis

The GO (http://geneontology.org, accessed on 9 October 2022) and KEGG pathway https://www.genome.jp/kegg/, accessed on 9 October 2022) databases were used to perform GO function annotations and metabolic pathway analysis, respectively, on the DEGs to obtain information regarding their associated biological functions and processes and their cellular locations.

### 2.5. Gene Ontology (GO) and Kyoto Encyclopedia of Genes and Genomes (KEGG) Enrichment Analyses

We utilized the Blast2GO function in R software to conduct functional enrichment analysis of the Differentially Expressed Genes (DEGs), and *p*-values were calculated using the hypergeometric distribution method to identify GO terms that were significantly enriched (*p* < 0.05) by DEGs [20]. KEGG pathway enrichment analysis of the DEGs was performed using ClusterProfiler (3.4.4) [21,22] to identify significantly enriched pathways. 

### 2.6. Quantitative Real-Time PCR (qPCR)

The expression patterns of selected DEGs in the liver were verified by qPCR using an Applied Biosystems 6300 RT-PCR system (Waltham, MA, USA). The target gene expression was determined via qPCR using a Roche LightCycler 480 II (Roche Diagnostics, Shanghai, China). The reaction volume for qPCR was 12.5 μL. The PCR conditions were as follows: initial denaturation at 95 °C for 30 s, followed by 40 cycles at 95 °C for 5 s and 60 °C for 30 s. Three samples for each temperature group were selected and the experiment was repeated three times for each sample to ensure accuracy (technical replicates). Six representative genes (including metabolism-, immunity-, and antioxidant-related genes) were selected for RNA-seq result verification. The primers were obtained using transcriptomics analysis. The primers are listed in Table 1; the amplification efficiency was greater than 90% for all primer pairs. EF-1α was chosen as the internal reference gene because it was not affected by temperature. The relative expression levels of the target genes relative to the control group were calculated using the 2^−ΔΔCT^ method [23].

### 2.7. Statistical Analyses

The data obtained from the experiments were analyzed using SPSS statistical software (26.0, IBM SPSS Inc., Chicago, IL, USA). All data are expressed as means ± standard errors (SEs). For comparisons of juvenile fish morphology and developmental stages between different temperature groups, significant differences between groups were determined using one-way analysis of variance (ANOVA), followed by Duncan’s multiple comparison test. Means were considered statistically significant at *p* < 0.05 [24].

## 3. Results

### 3.1. The Influence of Rearing Temperatures on the Development of P. teira

In the present study, we examined the effect of temperature on the development of the larvae and juveniles of *P. teira* (Table 2). The development from the fertilized egg to the mature *P. teira* includes three stages: the larval, juvenile, and young stages (Figure 1a–c). The larvae started to feed at approximately 3 days post-hatch (dph), and the yolk sac disappeared at approximately 5 dph. Overall, membrane rupture to juvenile development took 25 d. Larvae in the 27 °C group had the highest survival rate, followed by those in the 24 °C group. Additionally, *P. teira* reared at 24 °C had the lowest deformity rate, whereas those reared at 30 °C had the highest (Table 3). These results showed that the temperature extremes reduced the survival rate and increased the deformity rate of the fish.

### 3.2. Analysis of DEG at Different Breeding Temperatures

Differential expression analysis identified 5492, 6937, and 4246 DEGs in the 21 °C vs. 24 °C, 27 °C vs. 24 °C, and 30 °C vs. 24 °C comparisons, respectively. The highest number of DEGs was obtained in the 27 °C vs. 24 °C comparison, whereas the lowest was obtained in the 30 °C vs. 24 °C comparison. Additionally, the number of upregulated DEGs decreased gradually with increasing temperature, and the downregulated DEGs showed the opposite trend. Overall, the number of upregulated DEGs (10,568) was higher than that of downregulated DEGs (6107) (Table 4).

### 3.3. Transcriptome Sequencing and Raw Read Data Analysis

To obtain comprehensive transcriptomic information, de novo assembly was performed using the obtained raw sequencing reads. This allowed us to construct a transcriptome reference for subsequent analysis. A total of 12 high-quality cDNA libraries were constructed from the liver samples, and sequenced using the Nova Seq 6000 platform (Illumina) (Appendix A). In addition, there may be exogenous contaminants introduced into the sequencing sample, such as bacterial DNA, primer residues, etc. To ensure the accuracy and reliability of the analysis, those poor quality and contaminated readings were removed. Based on a preset quality threshold, reads below the threshold were filtered out or trimmed, and only higher-quality reads were retained for subsequent analysis. After removing low-quality and contaminated reads, a total of 20,576,389–29,293,587 clean reads were obtained from the liver samples, with Q20 and Q30 (Quality Score) > 97.3 and 93.7%, respectively. (Appendix A). Except for the 27 °C group, one sample was far away from the other two samples, which may be due to individual differences. The PCA analysis results of the remaining samples show that the samples in the same group basically clustered together, indicating that the repeatability of the samples can meet the needs of subsequent analysis (Figure 2).

### 3.4. GO and KEGG Annotation Analysis of DEGs

GO terms are largely classified into three categories: “Biological Process", “Molecular Function”, and “Cellular Component”. The top 50 enriched GO terms are listed in Figure 3. Additionally, KEGG pathway analysis was performed to identify pathways significantly affected by thermal stress, and the top 25 enriched pathways are shown in Figure 4. The DEGs were mainly enriched in five categories of KEGG pathways: metabolism, gene information, cellular processes, organic systems, and diseases. Specifically, DEGs in the 21 °C vs. 24 °C comparison were mainly enriched in pathways related to the metabolism of matter and energy, including pathways involving the proteasome, protein export, spliceosome, and valine, leucine, and isoleucine degradation, butanoate metabolism, the tricarboxylic acid cycle (TCA cycle), and tryptophan metabolism. DEGs in the 27 °C vs. 24 °C comparison were enriched in pathways associated with the digestion and absorption of proteins, including mineral absorption, cardiac muscle contraction, bile secretion, vasopressin-regulated water reabsorption, carbohydrate digestion and absorption, proximal tubular bicarbonate excretion, gastric acid secretion, and salivary secretion. DEGs in the 30 °C vs. 24 °C comparison were enriched in pathways associated with glucose and lipid metabolism, including glutathione metabolism, pentose and glucuronate exchange, steroid hormone biosynthesis, the proteasome pathway, cholesterol metabolism, fatty acid elongation, unsaturated fatty acid biosynthesis, fructose and mannose metabolism, cysteine and methionine metabolism, and the peroxisome proliferator-activated receptor alpha (PPAR) signaling pathway.

### 3.5. Validation of the Transcriptome Results

To validate the transcriptome results, the expression of catalase (*CAT*), interleukin-1 beta (*IL-1β*), glutathione peroxidase (*GPX*), heat shock protein 70 (*HSP70*), *PPAR*, superoxide dismutase (*SOD*), and six other genes were detected by qRT-PCR (Figure 5). The expression trends of the qRT-PCR and transcriptome results were identical, confirming the reliability of the RNA-seq data.

### 3.6. Peroxisome Proliferator-Activated Receptor Alpha (PPAR) Signaling Pathway

In the PPAR signaling pathway, VLDL/chylomicron in the liver transmits signals to FABP by upregulating (*p* < 0.05) FATP, and the activated FABP promotes the transcriptional expression of *PPAR* in the nucleus (Figure 6). Among the middle and downstream genes, the expression of gluconeogenesis-related genes (*CyK* and *PEPCK*) and lipid metabolism-related genes (*FABP1/4*, *ACS*, *CPT-1*, *CPT-2*, and *LCAD*) in the 21 °C vs. 24 °C comparison was upregulated (*p* < 0.05), whereas *CYP27* expression was downregulated (*p* < 0.05). In addition to the upregulation of gluconeogenesis and lipid metabolism-related genes, cell survival and ubiquitination-related genes were upregulated in the 27 °C vs. 24 °C comparison (*p* < 0.05). In the 30 °C vs. 24 °C comparison, FABP upregulated *PPAR*γ expression in the nucleus, and downregulated the expression of downstream genes, including *CyK, PGAR, SCP-X, PABP3*, and other related genes (*p* < 0.05).

### 3.7. The Expression Patterns of Glucose Metabolism-Related Genes

The expression of pyruvate carboxylase (*PC*), phosphoenolpyruvate carboxykinase (*PCK*), citrate synthase (*CS*), pyruvate kinase (*PK*), lactate dehydrogenase (*LDH*), glucokinase (*GK*), hexokinase (*HK*), phosphofructokinase (*PFK*), and glucose transporter 1 (*GLUT1*) in the 21 and 27 °C groups increased (*p* < 0.05) compared to that in the control group (Figure 7). However, the genes had significantly lower (*p* < 0.05) expression patterns in the 30 °C group compared with that in the 24 °C group.

### 3.8. Tricarboxylic Acid Cycle (TCA Cycle)

In the present study, 28 genes (including *CS*, *IDH*, *MDH*, *FH*, and *LSC*) associated with substance and energy metabolism and the TCA cycle pathway were upregulated in the 21 °C group compared with that in the 24 °C group (*p* < 0.05, Figure 8). Additionally, the expression of 23 genes in the 27 °C group increased compared with that in the 24 °C group (*p* < 0.05). However, the expression of 20 genes in the 30 °C group decreased compared with that in the control group (*p* < 0.05).

## 4. Discussion

### 4.1. Influence of Temperature on Larval and Juvenile Growth and Development

The larval and juvenile stages are two important stages of postembryonic development in the early life of fish. The morphology and digestive organs of the larvae are in the fast developmental phase and their emergence and development rate are closely related to the breeding environment [25,26]. No effect of temperature on the early developmental stages of *P. teira* or other batfish species has been found in the existing reports. In the present study, thermal stress significantly influenced the development and survival of the larvae and juveniles of *Platax teira*. Additionally, the yolk sac consumption rate was faster in the high temperature groups (27 °C and 30 °C), with early onset of exogenous feeding, indicating that high temperature promoted exogenous feeding. Moreover, higher temperatures promoted the development of scales in the juveniles, as evidenced by the rapid development of scales in juveniles in the high temperature groups (27 °C and 30 °C). In this experiment, the larvae of *P. teira* experienced 25 days of development to the juvenile stage at 27 °C and 30 °C. However, the mortality and deformity rates of the 30 °C group were much higher than those of the 27 °C group. These results showed that increasing the temperature within the acceptable threshold could promote organ development and fish growth; however, extremely high temperatures could be detrimental to growth and development [27]. Similarly, Cavrois-Rogacki et al. [28] reported that exposure to optimal temperatures can improve the feeding rhythm and growth rate of fish, and can shorten the breeding period of juvenile *Labrus bergylta*.

Different species of fish have different temperature ranges, which are related to their unique genetic characteristics and long-term adaptation to the environment [29]. In the present study, the feeding ability and survival rate of *P. teira* larvae and juveniles exposed to low-temperature stress (21 °C) decreased compared with that in the other groups. Moreover, in a study of salmon (*Oncorhynchus*) by Jin et al., low temperature was shown to inhibit the development of various organs necessary for the complete absorption of exogenous nutrients, which can cause larvae death [30]. However, larvae in the high-temperature group (30 °C) had a lower survival rate and higher abnormality rate than those in the 24 and 27 °C groups. This is consistent with Lee et al.’s study on the growth of cultured sablefish (*Anoplopoma fimbria*) juveniles at high temperatures [31]. Extreme temperatures can negatively affect the survival and development of larvae and juveniles. Additionally, larvae death peaked approximately one week after fin emergence, whereas the lowest death rate was observed in the juvenile stage, indicating that the larval stage was more sensitive to temperature variations. The larval stage is a transitional period, during which the organ systems are under development, which could be responsible for the higher susceptibility of larvae to temperature variations.

### 4.2. GO and KEGG Analyses of DEGs

The liver is an important organ for metabolism in fish and plays a vital role in cell metabolism, biosynthesis, and detoxification [32]. In the present study, RNA-seq was performed to elucidate the effect of breeding temperature (21, 24, 27, and 30 °C) on the transcriptome of the liver of *P. teira*. The expression patterns of several genes were significantly affected by thermal stress, indicating that thermal stress response in *P. teira* is regulated by multiple genes. Under thermal stress conditions, *P. teira* may regulate body functions to survive by controlling the expression of several genes, which was consistent with results in zebrafish [33], flounder [34], and large yellow croaker [9]. Additionally, the number of DEGs between the 21 °C and 24 °C control groups was significantly higher than that between the 30 °C and 24 °C groups, indicating that low-temperature stress had a larger effect on *P. teira* than high-temperature stress. GO function enrichment analysis showed that the DEGs were mainly enriched in metabolic processes, cellular processed, single-organism processed, catalytic activity, cell part, membrane, and membrane part. This may be due to changes in the cellular structure and function of *P. teira* under temperature stress, which may have activated several structural proteins and enzymes to improve metabolism in response to environmental stress [35]. Furthermore, the KEGG pathway enrichment analysis showed that the DEGs were significantly enriched in metabolism pathways, including butanoate, tryptophan, galactose, and nitrogen metabolism and the TCA cycle. These results indicate that the TCA cycle is involved in the synthesis and catabolism of major nutrients, and transports metabolites into the cytoplasm to provide raw materials for the synthesis of related functional molecules [36].

### 4.3. The Effect of Temperature on the PPAR Signaling Pathway

The PPAR signaling pathway plays an important role in the secretion of adipokines and the regulation of adipocyte differentiation [37]. PPAR has three subtypes, namely PPARα, PPARβ, and PPARγ [38]. These three isoforms are endogenous ligands with high affinity, and act as lipid sensors that specifically bind to target genes and initiate transcriptional processes to mediate metabolic regulation [39]. However, the three subtypes have different genetic codes and different tissue distributions. In the present study, the DEGs were highly enriched in the PPAR signaling pathway in response to thermal stress, which was consistent with the findings in gilthead seabream [40] and large yellow croaker [9]. These results indicate that the PPAR signaling pathway plays a key role in regulating thermal stress responses in several fish species. Under low-temperature stress, very-low-density lipoprotein (VLDL/chylomicron) in the liver of *P. teira* transmits signals to FABP through FATP, and the activated FABP promotes the transcription and expression of PPARγ in the nucleus. Additionally, thermal stress upregulated the expression of gluconeogenesis-related genes (CyK and PEPCK) and lipid metabolism genes (FABP1/4, ACS, CPT-1, CPT-2, and LCAD), but downregulated *CYP27* expression, indicating that *P. teira* responds to the low-temperature stress by increasing glycolysis and lipid metabolism. This further confirms that, under low temperature conditions, larvae obtain more energy to cope with environmental pressure to maintain physiological balance, resulting in slow growth and an increased deformity rate. Tsai et al. [41] reported that PPARγ is closely related to fat anabolism in cobia, and its expression level is significantly correlated with tissue fat content. Additionally, CPT-1 and CPT-2 are two important members of the fatty acylcarnitine transfer system, and are also key enzymes in fatty acid β-oxidation [42]. These findings indicate that *P. teira* promotes lipid anabolism through the PPAR signaling pathway and enhances lipid catabolism under low-temperature stress.

Furthermore, PPARγ is critical in controlling genes involved in glucose homeostasis by regulating its target genes, including *PCK* and *GK*. PCK is the rate-limiting enzyme of gluconeogenesis and can cause an increase in the accumulation of triglycerides [43]. An increase in GK can induce glycolysis and increase malonyl-CoA production, which can promote triglyceride accumulation by inducing fatty acid synthesis and inhibiting fatty acid oxidation [44]. In the present study, the expression of key enzymes of PPARγ, including CyK, PGAR, SCP-X, and PABP3, decreased with increasing temperature, which decreased lipid metabolism and gluconeogenesis. These findings indicate that high temperature does not support carbohydrate metabolism and fat accumulation in swallowtail. Similarly, Li et al. [45] reported that high temperatures are not conducive for the uptake of exogenous substances in rainbow trout and reduce the metabolic rate and activities of metabolic enzymes.

### 4.4. The Effects of Temperature on Glucose Metabolism and TCA Cycle

Carbohydrates provide energy for metabolism in fish, and are involved in maintaining normal physiological functions [46]. A previous study showed warm-water fish have a higher carbohydrate utilization than cold-water fish, indicating that carbohydrate utilization by fish is influenced by temperature [47]. In the present study, low- (21 °C) and high-temperature stress (30 °C) negatively affected the growth and development of *P. teira*. Aquatic organisms resist and adapt to unfavorable environments by regulating energy metabolism; in particular, fish adopt different physiological and biochemical strategies to cope with thermal stress, and changes in energy metabolism in fish are usually measured by the level of glucose metabolism [48]. This distinct pattern in comparison to lipid and protein regulation could be related to the importance of carbohydrates as an immediate fuel source and in terms of maintaining blood glucose levels. It is possible that glycogen metabolism-related gene upregulation was ‘saved’ for anaerobic glycolysis during burst activity, which is essential for fish to catch prey or to escape predators, generally termed the fight or flight response. It has been speculated that there are two factors responsible for the upregulation of carbohydrate genes in fish due to temperature: (1) temperature accelerates the digestion and absorption of carbohydrates by fish; and (2) temperature increases energy metabolism in fish, which in turn, enhances the ability of the body to oxidize and decompose sugar for energy. In the present study, breeding temperature significantly affected the glycogen content in the liver of *P. teira*. The liver’s ability to absorb carbohydrates has been speculated to increase with increasing environmental temperature, which can promote the accumulation of hepatic glycogen. Fish exposed to low-temperature stress (21 °C) exhibited an increase in carbohydrate utilization to generate energy to cope with the environment, which was consistent with the second effect of temperature on carbohydrate utilization.

PFK and PK are rate-limiting enzymes in glycolysis and are highly expressed in fish liver [49]. In the present study, HK and PFK expression patterns in the liver of *P. teira* exhibited a quadratic pattern, increasing initially with the increase in temperature, and then decreasing at higher temperatures. The reason may be that the increase in water temperature considerably increased energy consumption and metabolism in the body. Moreover, HK and PFK are the key enzymes in the first stage of glycolysis, which is an energy-consuming stage [50]. Therefore, we speculated that the insufficient supply of ATP in fish under high-temperature stress reduced the concentration of glucose, resulting in a decrease in the expression of HK and PFK in the high-temperature group (30 °C). An increase in environmental temperature within the acceptable threshold has been shown to increase sugar absorption and digestion in fish, indicating that increased temperature promotes the expression of gluconeogenesis-related genes and glycogen accumulation [51]. However, extremely high-temperature conditions can suppress sugar digestion and absorption in fish, which can inhibit sugar metabolism [52]. Under lower temperature conditions, fish may increase glycolysis and energy to resist the low-temperature environment. 

The TCA cycle is an important pathway for the complete oxidation of proteins, sugars, and fats, and it is also a key hub connecting the three major substances [53]. Acetyl-CoA is the intersection between lipid metabolism and the TCA cycle in organisms. Free fatty acids are decomposed by β-oxidation, and the resulting acetyl-CoA can enter the TCA cycle. The citric acid in the TCA cycle is catalyzed by ATP citrate lyase to generate acetyl-CoA, and organisms can also use acetyl-CoA in the TCA cycle for fatty acid synthesis [54]. PK and CS are important indicators of changes in energy metabolism in fish, and the process of the PK and CS reactions can release energy. In the present study, PK and CS expression patterns were consistent with the expression patterns of glycolysis-related genes. Specifically, PK and CS expression increased significantly under low-temperature conditions, which is necessary for increased energy metabolism to resist environmental stress. However, Wen et al. [55] reported that prolonged exposure to cold stress can weaken energy metabolism in discus fish (*Symphysodon aequifasciatus*). Moreover, differential expression analysis showed an increase in the expression of genes associated with the TCA cycle in fish exposed to cold stress, including *CS*, *PC*, *IDH*, *SDH*, and *ACO*. An improvement in the conversion efficiency of the TCA cycle under low-temperature conditions is beneficial for the regeneration of important nutrients in the body. However, extreme temperatures can inhibit energy metabolism in fish, resulting in oxidative stress and energy utilization disorders in organisms. Additionally, these findings confirm the key role of the TCA cycle in thermal stress responses in *P. teira*.

## 5. Conclusions

Our findings showed that PPAR signaling, glucose metabolism, and the TCA cycle play important roles in thermal stress responses in *Platax teira*. Additionally, exposure to low-temperature environments improved energy, lipid, and glucose metabolism. In contrast, high temperatures above the optimal range decreased energy, lipid, and glucose metabolism, thereby inducing oxidative stress and energy loss. Overall, a breeding temperature of 24–27 °C supported optimal larvae growth and development compared with the other breeding temperatures, and should be adopted in *P. teira* breeding. Through transcriptomics, the regulatory mechanism of larval development in *P. teira* under different growth temperatures was elucidated, with the goal of establishing a theoretical basis for industrial breeding.

## Figures and Tables

**Figure 1 biology-12-01161-f001:**
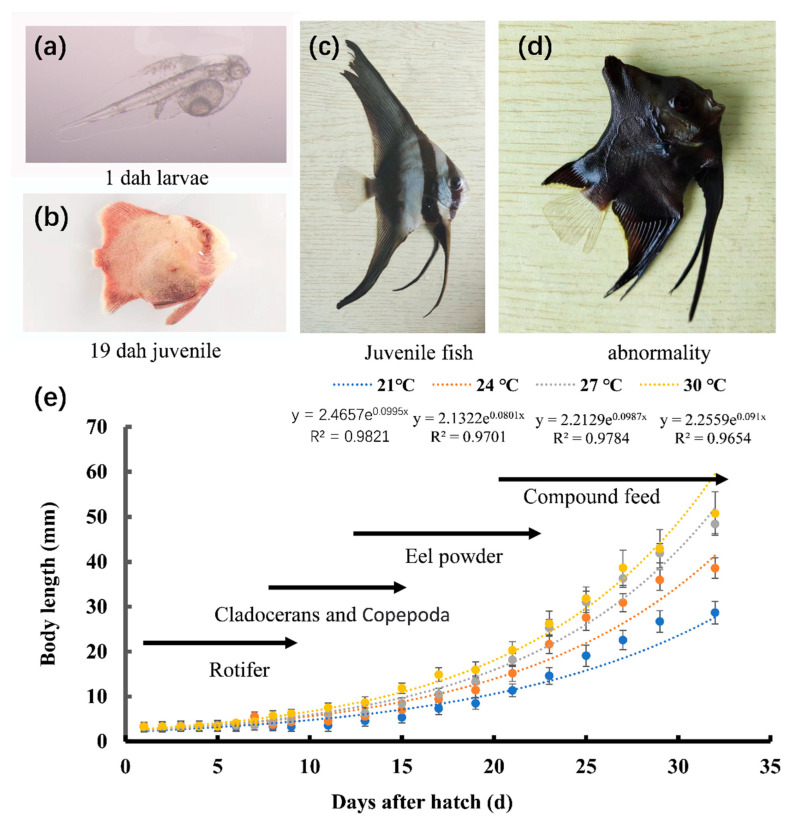
Feeding strategies and growth curves of *P. teira* larvae and juveniles during aquaculture. Images of (**a**) 1 dph larvae; (**b**) 19 dph juveniles; (**c**) juveniles; (**d**) abnormal individuals. (**e**) Feeding strategy and growth curve.

**Figure 2 biology-12-01161-f002:**
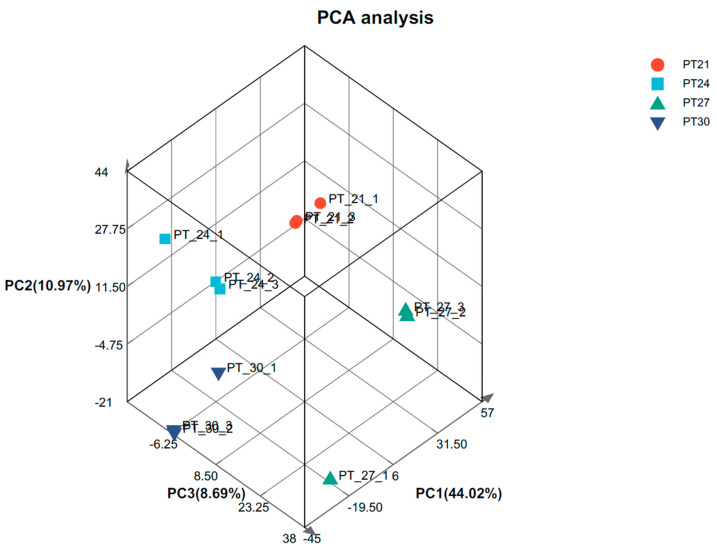
Gene expression PCA analysis. Note: After dimensionality reduction analysis, there are relative coordinate points on the principal component. The distance of each sample point represents the distance of the sample. The closer the distance, the higher the similarity between samples.

**Figure 3 biology-12-01161-f003:**
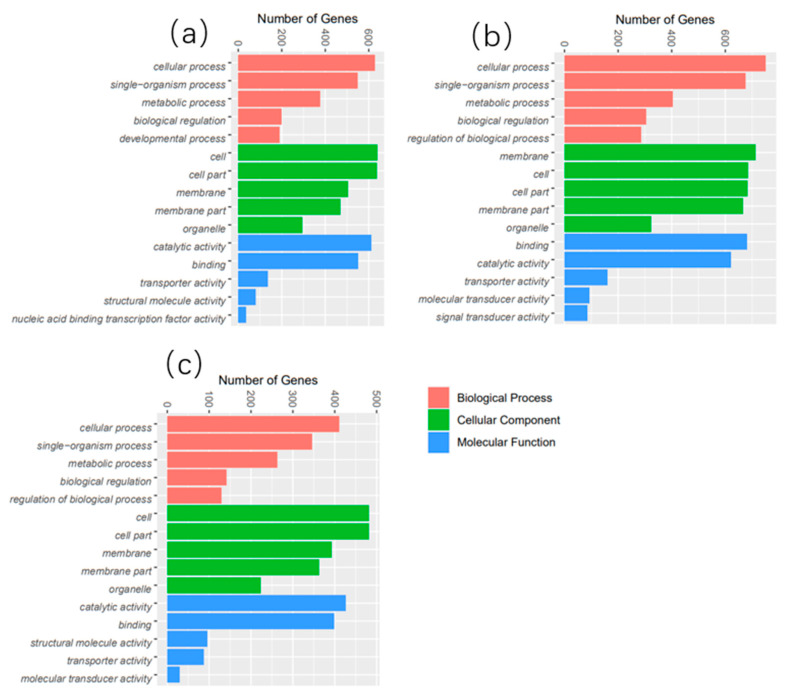
The top 50 most abundant GO terms in the liver of *P. teira* cultured under different temperature conditions. (**a**) 21 °C vs. 24 °C; (**b**) 27 °C vs. 24 °C; (**c**) 30 °C vs. 24 °C.

**Figure 4 biology-12-01161-f004:**
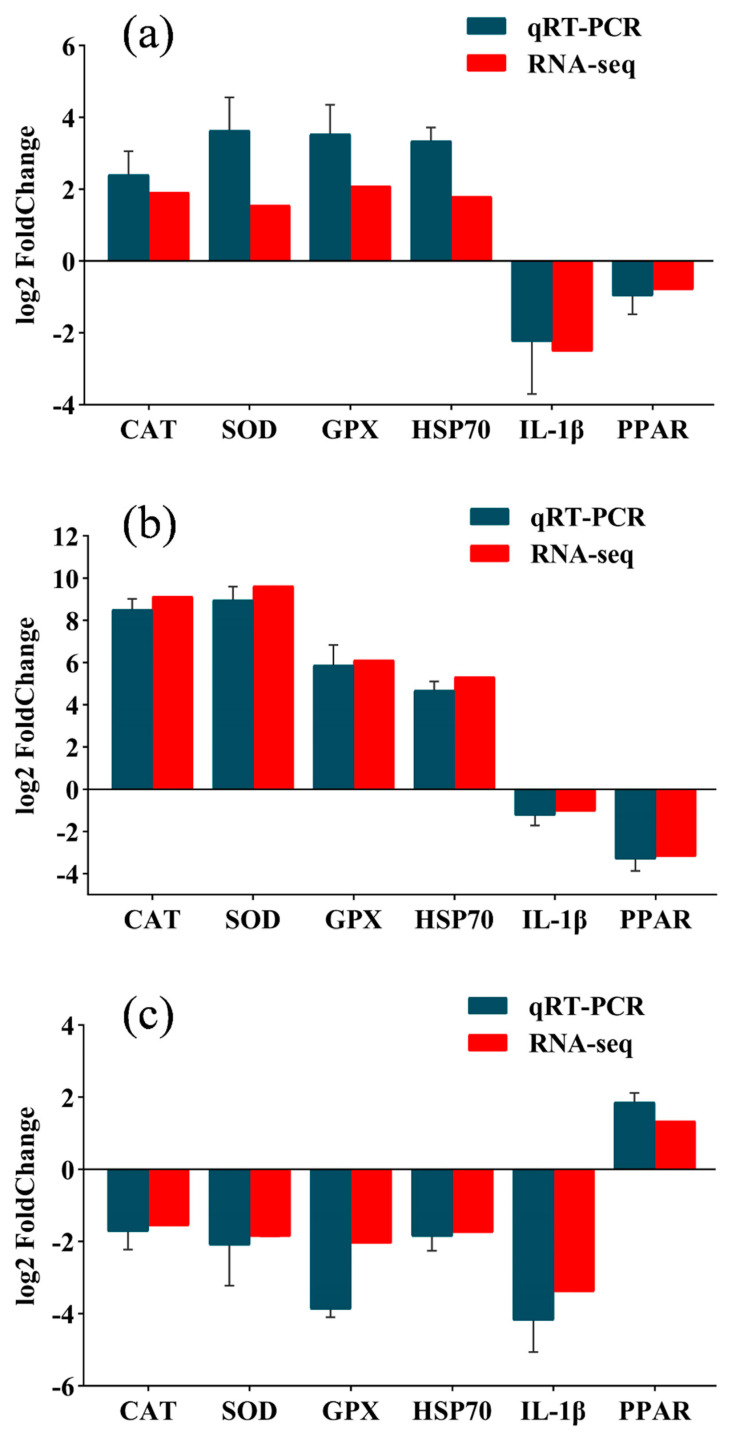
Verification of transcriptome results. (**a**) 21 °C vs. 24 °C; (**b**) 27 °C vs. 24 °C; (**c**) 30 °C vs. 24 °C.

**Figure 5 biology-12-01161-f005:**
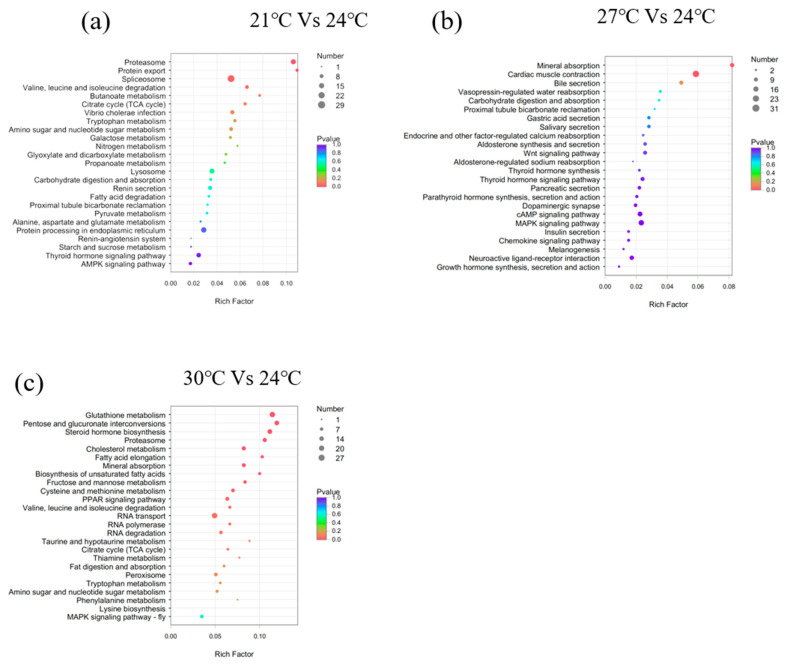
Bubble chart of the top 25 enriched KEGG pathways of the DEGs. (**a**) 21 °C vs. 24 °C; (**b**) 27 °C vs. 24 °C; (**c**) 30 °C vs. 24 °C. Note: The vertical axis represents the pathway categories, and the horizontal axis shows the gene ratio. The point size reflects the number of DEGs in the enriched KEGG pathway. The point color reflects different Q values as indicated on the right.

**Figure 6 biology-12-01161-f006:**
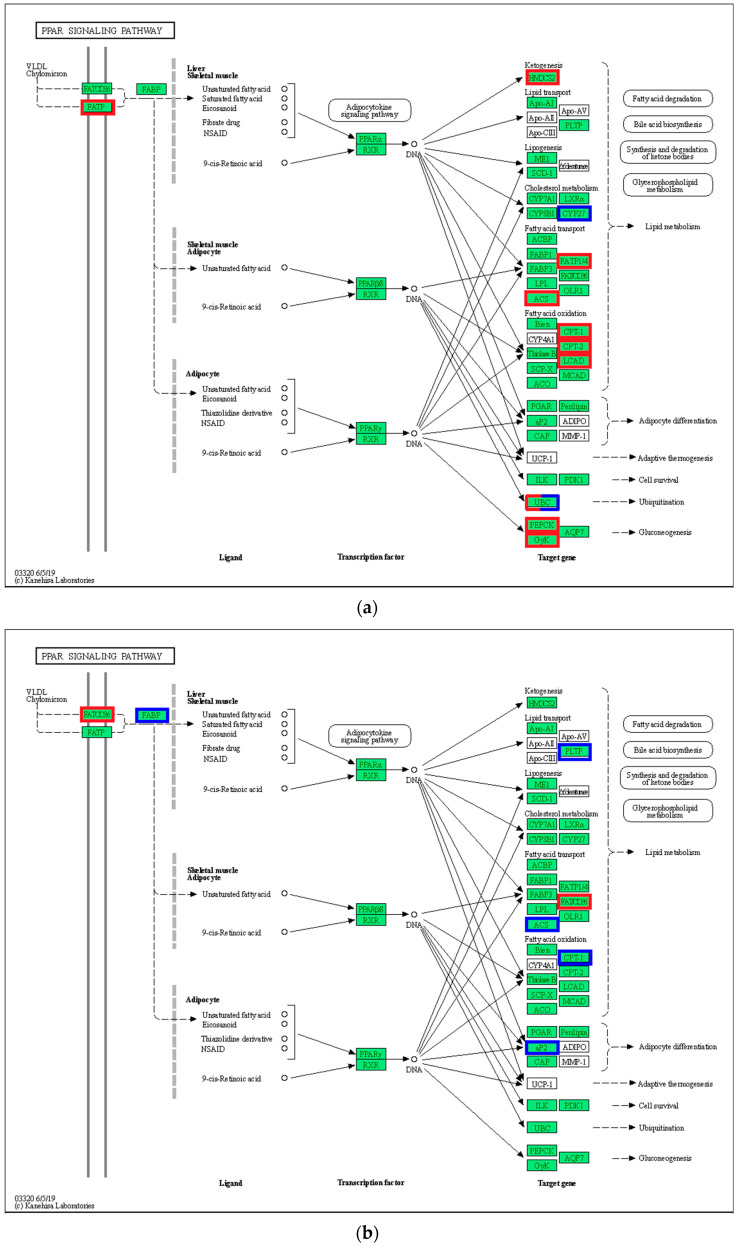
PPAR signaling pathway. (**a**) 21 °C vs. 24 °C; (**b**) 27 °C vs. 24 °C; (**c**) 30 °C vs. 24 °C. Note: red represents up-regulated gene, blue represents down-regulated gene.

**Figure 7 biology-12-01161-f007:**
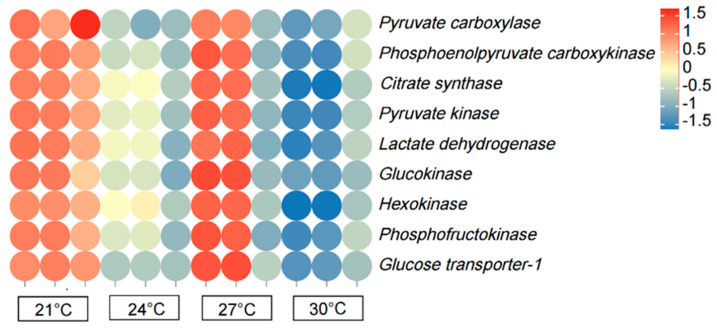
Heat map illustrating the row-normalized log10(FPKM) values of genes related to glucose metabolism in the liver of *P. teira* at different temperatures. Note: Heatmaps represents the expression levels of genes, rather than specifically highlighting the differentially expressed genes (DEGs). The point color reflects different Q values.

**Figure 8 biology-12-01161-f008:**
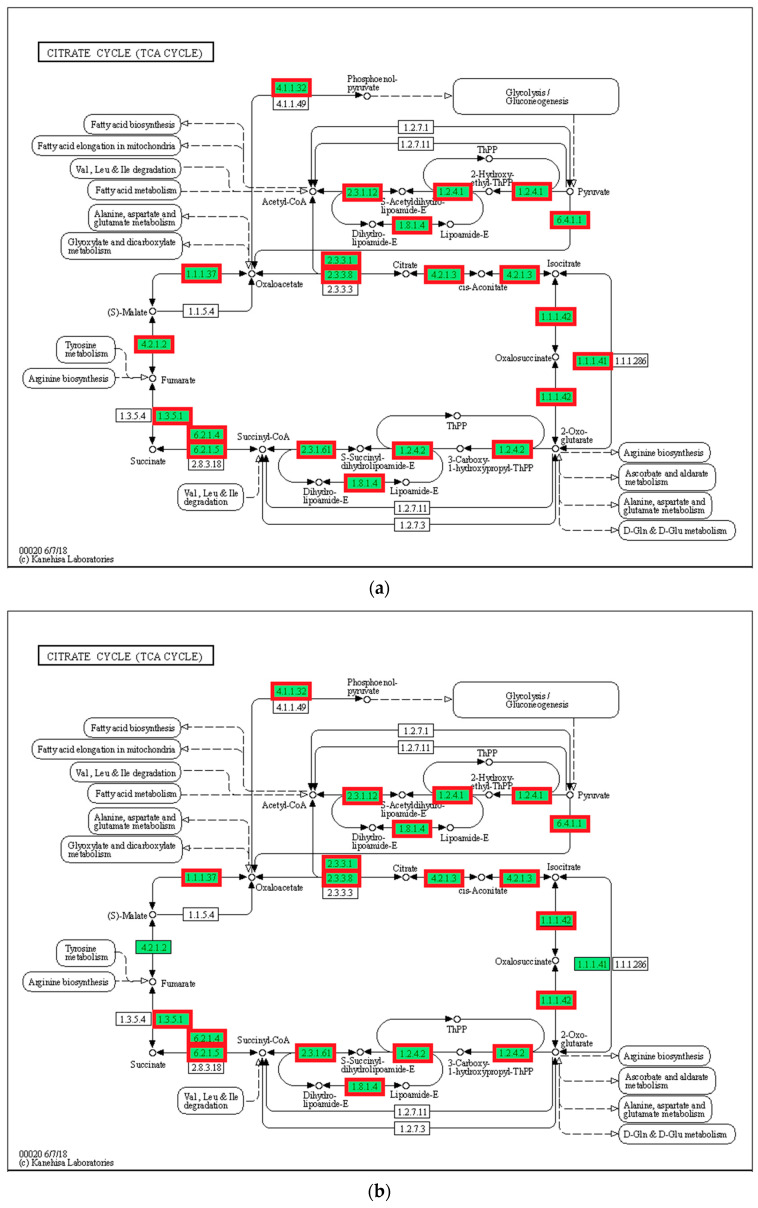
TCA cycle. (**a**) 21 °C vs. 24 °C; (**b**) 27 °C vs. 24 °C; (**c**) 30 °C vs. 24 °C. Note: red represents up-regulated genes, blue represents down-regulated genes.

**Table 1 biology-12-01161-t001:** Primers used for mRNA expression analysis.

Primer Name	Sequence (5′-3′)	Length	Amplification Target
CAT	F: TCCCGTTTCTGGCGATGTT R: TCCTGGATGAAGGGTTGTGC	19 20	qRT-PCR
IL-1β	F: AGCAGAGGGCAACAACAAG R: TCCCCACAGGTAGAATCACA	19 20	qRT-PCR
GPX	F: TGGACAGCGTATCCGACTT R: GAGCGATGCGTTCTTCTTTA	19 20	qRT-PCR
HSP70	F: TTCAAGGTGCTGGGAGATG R: TGCGTCTTTAGTCGCCTGT	19 19	qRT-PCR
PPAR	F: CACTGTTTCTGGCTGTCATAATC R: ACGCAGGTCGGTCATTTTC	23 19	qRT-PCR
SOD	F: TTCAGGCTCAATCAATGGTC R: TTCCGAAGGGGTTGTAGTG	20 19	qRT-PCR
EF	F: AAGCCAGGTATGGTTGTCAACTTT	24	qRT-PCR
	R: CGTGGTGCATCTCCACAGACT	21	

**Table 2 biology-12-01161-t002:** The effects of temperature on the development of *Platax teira* larvae and juveniles.

Developmental Stage	Days (d)			
	21 °C	24 °C	27 °C	30 °C
Larvae feeding	3 ^b^	3 ^b^	3 ^b^	2.5 ^a^
Disappearance of the yolk sac	5 ^b^	5 ^b^	4 ^a^	4 ^a^
Appearance of scales	24 ^d^	20 ^c^	19 ^b, c^	18 ^a, b^
Appearance of intact scales	32 ^c^	27 ^b^	25 ^a^	25 ^a^

Note: There are significant differences between different letters. Ten fish from each temperature group were selected to determine the developmental stage.

**Table 3 biology-12-01161-t003:** Effects of different temperatures on *Platax teira* Survival rate and Abnormality rate.

	Different Temperature Groups			
	21 °C	24 °C	27 °C	30 °C
Survival rate (%)	46 ^a^	63 ^c^	67 ^d^	52 ^b^
Abnormality rate (%)	8 ^c^	5 ^b^	6 ^a^	12 ^d^

Note: There are significant differences between different letters. For each temperature group, 50 fish were selected at a time to determine the Survival rate and Abnormality rate.

**Table 4 biology-12-01161-t004:** Statistics of differentially expressed genes (DEGs) under different temperatures.

Time	Up-Regulated (%)	Down-Regulated (%)	Total
21 °C vs. 24 °C	5087 (48.1)	405 (6.6)	5492
27 °C vs. 24 °C	4900 (46.4)	2037 (33.4)	6937
30 °C vs. 24 °C	581 (5.5)	3665 (60.0)	4246
Total	10,568	6107	16,675

## Data Availability

All raw reads were submitted to the Sequence Read Archive (SRA) (accession: PRJNA872865).

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
