# Peer review of "Transcriptomics Reveal the Effects of Breeding Temperature on Growth and Metabolism in the Early Developmental Stage of Platax teira"

_biology, 2023, doi:10.3390/biology12091161_

Round 1

Reviewer 1 Report

This study investigates the effects of breeding temperature on the development and transcriptomic response of Platax teira. The study is relevant as it proposed to elucidate thermal stress mechanisms through a transcriptomic analysis focusing on energy, lipid and glucose metabolism pathways. The originality here is to use a transcriptomic analysis with non-model organisms as the physiological/breeding impact of temperature modulation is already well-known in fish at the individual level. However, some clarification needs to be made regarding the methods used.

Introduction

Line 74: remove “its meat is delicious” as it is not relevant in a scientific paper. Authors could provide more information about the economic value of the species to emphasize the relevance of testing breeding temperature.

Materials and methods

Line 90: should read “…the eggs were artificially fertilized and transferred…”

Line 109: should read “…three times a day to ensure sufficient…”

In the 2.1 section, the authors write that the eggs were incubated at 27°C. However, the control temperature was set to 24°C for larvae during the experiment. Is there a reason why the optimal temperature was changed between the stages? Furthermore, it is said that the temperature adjustment was performed from 24°C while the incubation was at 27°C. Please provide more information about the rearing requirements for this species as it could be difficult for not familiarized readers.

L111: Please provide details about the morphological and developmental endpoints observed

The transcriptomic workflow needs some clarifications and more details, especially concerning the read alignment, did the authors perform a de novo assembly? It is not clear for a reader not familiar with non-model organisms lacking a genome.

L145: add a reference for HTSeq

L137: should read “…genes with absolute log2 fold change > 1…” and if a p-value adjustment was applied please change p-vale for adjusted p-value plus the type of correction.

L145: if the topGo package was used please change “function” to “package”. Furthermore, the reference [19] returns to a paper about the Blast2Go suite. If the Blast2GO suite was used instead of the topGO package in R, please refer to the Blast2GO suite in the section. Otherwise, use a more appropriate reference for the topGO package.

In the 2.6. section, please provide more information about the data obtained from the experiments that were analyzed. The readers could guess that the authors refer to the morphological and developmental endpoints but it is worth clarifying. Also, were they all tested by ANOVA? Table 3 also refers to percentages.

Results

L175: The sentence about the 19°C exposure could be removed as it was already said in the M&M section. Otherwise, please provide supplementary data.

L177: Did the authors could provide data regarding the fact that larvae were most active under high temperatures?

L180: it is said that fish reared at 27°C exhibit the lowest malformation rate but the higher percentage of malformation was at 24°C in Table 3.

In sections 3.2. ; 3.4. and 3.6, the authors said that the DEGs were identified by dividing the control group (24°C) by other groups (i.e. 24 vs 21 / 24 vs 27 / 24 vs 30). Usually for the DEGs analysis the treated groups are divided by the control group. Then, in all figure’s captions, this ratio is inverted to match a more classical way of analyzing the DEGs data (i.e. 21 vs 24 / 27 vs 24 / 30 vs 24). If this is a typo in the text, please correct it as even if this inversion did not change the results per se, the interpretation could be different if based on the up- or down-regulation of genes.

L186: It is said that the highest number of DEGs was obtained in the 21°C group but according to Table 2 it is in the 27°C group.

The 3.3 section should be added to the M&M just after the sequencing as it is part of the quality control to perform subsequent data analysis (DEG and enrichment).

Please provide the entire list of significant DEGs, GO and KEGG annotations in supplementary data.

Discussion

L363: Is there data on the glycogen content in the present study? If the authors refer to the potential modification of glycogen content based on genes dysregulation and pathways highlighted it should be said as so. Idem for carbohydrate utilization line 366.

General comments

-          Figure 3 and 4 captions are inverted

-          Homogenization is needed for the x-axis in the Figure 3C

-          Is it really the differential expression mapped on the heat map as it also provides data for the control group? And please provide a caption for the legend.

-          Table 3 is cited before Table 2 in the text please correct the tables numerations

-          Table 3 is not clear, there is an incomplete note and data in % cross data in days. Please add the SD and the number of individuals used

-          No cross-comparisons were made between DEGs or enriched pathways from the 21, 27 and 30°C groups to highlight common dysregulated genes or pathways. Even if common pathways were highlighted by authors focusing on 3 main pathways this analysis could be relevant in RNAseq analysis.

Author Response

We gratefully thank the editor and all reviewers for their time spend making their constructive remarks and valuable suggestions, which has significantly improved the quality of the manuscript and has enabled us to improve the manuscript. Each suggested revision and comment brought forward by the reviewers was accurately incorporated and considered. Below the comments of the reviewers are responses point by point, and the revisions are indicated. Modifications are marked with color.

Introduction

Line 74: remove “its meat is delicious” as it is not relevant in a scientific paper. Authors could provide more information about the economic value of the species to emphasize the relevance of testing breeding temperature.

Response: Thanks for the suggestions. According to the comments from you, we have already corrected this error.

Materials and methods

Line 90: should read “…the eggs were artificially fertilized and transferred…”

Response: Thanks for the suggestions. According to the comments from you, we have already corrected this error.

Line 109: should read “…three times a day to ensure sufficient…”

Response: Thanks for the suggestions. According to the comments from you, we have already corrected this error.

In the 2.1 section, the authors write that the eggs were incubated at 27°C. However, the control temperature was set to 24°C for larvae during the experiment. Is there a reason why the optimal temperature was changed between the stages? Furthermore, it is said that the temperature adjustment was performed from 24°C while the incubation was at 27°C. Please provide more information about the rearing requirements for this species as it could be difficult for not familiarized readers.

Response: Thank you for your valuable comments, which greatly helped us with the manuscript. The brood stock is spawned in Hainan, the natural spawning water temperature is 27°C, and it is airlifted to Shenzhen, where the natural incubation temperature is 24°C. This section has been revised to clarify the experimental temperature more clearly.

L111: Please provide details about the morphological and developmental endpoints observed

Response: Thanks for the suggestions. Following your suggestion, we have added details of the observed morphological and developmental endpoints as follows:

The newly hatched larvae are transparent in appearance, with a considerable distribution of branchiostegal rays on the body surface. They sequentially go through the fry stage and the juvenile stage before entering the young fish stage. During the young fish stage, the fin spines and fin rays have fully developed, and a characteristic feature includes three black stripes on the head, trunk, and tail, while the body appears silver-white.

The transcriptomic workflow needs some clarifications and more details, especially concerning the read alignment, did the authors perform a de novo assembly? It is not clear for a reader not familiar with non-model organisms lacking a genome.

Response: Thanks for the suggestions. According to the comments from you, This part is supplemented in 2.4 Bioinformatics Analysis.

L145: add a reference for HTSeq

Response: Thanks for the suggestions. According to the comments from you, we added references as follows: Simon A, Theodor P P, Wolfgang H. HTSeq--a Python framework to work with high-throughput sequencing data. (2015). Bi-oinformatics (Oxford, England), 31(2). https://doi.org/10.1093/bioinformatics/btu638

L137: should read “…genes with absolute log2 fold change > 1…” and if a p-value adjustment was applied please change p-vale for adjusted p-value plus the type of correction.

Response: Thanks for the suggestions. According to the comments from you, we have already corrected this error.

L145: if the topGo package was used please change “function” to “package”. Furthermore, the reference [19] returns to a paper about the Blast2Go suite. If the Blast2GO suite was used instead of the topGO package in R, please refer to the Blast2GO suite in the section. Otherwise, use a more appropriate reference for the topGO package.

Response: Thanks for the suggestions.This section is a typo, and this study used the Blast2GO suite instead of the topGO package in R.

In the 2.6. section, please provide more information about the data obtained from the experiments that were analyzed. The readers could guess that the authors refer to the morphological and developmental endpoints but it is worth clarifying. Also, were they all tested by ANOVA? Table 3 also refers to percentages.

Response: Thanks for the suggestions. According to the comments from you, we have already corrected this error.

Results

L175: The sentence about the 19°C exposure could be removed as it was already said in the M&M section. Otherwise, please provide supplementary data.

Response: Thanks for the suggestions. According to the comments from you, we have already corrected this error.

L177: Did the authors could provide data regarding the fact that larvae were most active under high temperatures?

Response: Thanks for the suggestions. This has been supplemented in the manuscript.

L180: it is said that fish reared at 27°C exhibit the lowest malformation rate but the higher percentage of malformation was at 24°C in Table 3.

Response: Thanks for the suggestions. Due to the inconvenience caused by the writing error, the deformity rate of larvae is lower at 24°C.

In sections 3.2. ; 3.4. and 3.6, the authors said that the DEGs were identified by dividing the control group (24°C) by other groups (i.e. 24 vs 21 / 24 vs 27 / 24 vs 30). Usually for the DEGs analysis the treated groups are divided by the control group. Then, in all figure’s captions, this ratio is inverted to match a more classical way of analyzing the DEGs data (i.e. 21 vs 24 / 27 vs 24 / 30 vs 24). If this is a typo in the text, please correct it as even if this inversion did not change the results per se, the interpretation could be different if based on the up- or down-regulation of genes.

Response: Thanks for the suggestions. According to the comments from you, we have already corrected this error. Sorry for the inconvenience caused by typos.

L186: It is said that the highest number of DEGs was obtained in the 21°C group but according to Table 2 it is in the 27°C group.

Response: Thanks for the suggestions. Due to the inconvenience caused by the writing error. The highest number of DEGs was obtained in the 27 ℃ vs 24 ℃ group.

The 3.3 section should be added to the M&M just after the sequencing as it is part of the quality control to perform subsequent data analysis (DEG and enrichment).

Please provide the entire list of significant DEGs, GO and KEGG annotations in supplementary data.

Response: Thanks for the suggestions. The required data for this study, which involves the analysis of a de novo transcriptome, have all been presented.

Discussion

L363: Is there data on the glycogen content in the present study? If the authors refer to the potential modification of glycogen content based on genes dysregulation and pathways highlighted it should be said as so. Idem for carbohydrate utilization line 366.

Response: Thanks for the suggestions. This content has been revised in the manuscript.

General comments

Figure 3 and 4 captions are inverted

Response: Thanks for the suggestions. According to the comments from you, we have already corrected this error.

Homogenization is needed for the x-axis in the Figure 3C

Response: Thanks for the suggestions. According to the comments from you, we have already corrected this error.

Is it really the differential expression mapped on the heat map as it also provides data for the control group? And please provide a caption for the legend.

Response: Thanks for the suggestions. The heat map reflects the actual experimental data. The fluctuations observed in the control group data may be attributed to the experimental conditions that simulated practical aquaculture practices. This approach is more conducive to gaining insights into the actual scenario.

Table 3 is cited before Table 2 in the text please correct the tables numerations

Response: Thanks for the suggestions. According to the comments from you, we have already corrected this error.

Table 3 is not clear, there is an incomplete note and data in % cross data in days. Please add the SD and the number of individuals used

Response: Thanks for the suggestions. We revised the table format. Each temperature group was selected 10 fish at a time to determine the developmental stage.

No cross-comparisons were made between DEGs or enriched pathways from the 21, 27 and 30°C groups to highlight common dysregulated genes or pathways. Even if common pathways were highlighted by authors focusing on 3 main pathways this analysis could be relevant in RNAseq analysis.

Response: Thanks for the suggestions. Due to the prior utilization of some experimental data by other articles, this study did not involve the cross-comparison of differentially expressed genes (DEGs) or enrichment pathways in the 21°C, 27°C, and 30°C groups.

Reviewer 2 Report

Line 74- The “meat is delicious” is a subjective statement and opinion, should be removed.

Lines 74-76- This statement should be referenced.

Line 76- What is considered to be extreme weather?

Line 87- A further description of “nutrient fortification” is needed for clarity.

Line 89- How many eggs? How many fertilized? How were they collected? Were the eggs pooled?

Line 99- Were these static or flow through systems?

Line 105- At what stage hatch?

Line 109- What type of “bait” and how much were they allowed to eat? Was it calculated per body weight?

Lines 109-110- Not clear what is meant by “ensure sufficient food and satiation feeding.”

Line 111- Not clear what was assessed morphologically or what developmental changes consisted of. More detail needed.

Line 113- This suggests that n = 1, when in fact n = 3. Were three separate fish pooled together to make 1 sample for RNA seq, so using 9 fish total?

Line 114- Freezer

What kits were used for library prep?

Were these single or paired end? How many bp?

Line 131- What was considered and acceptable RIN?

Line 132- Sequences need to be made publicly available.

Section 2.3: How were raw reads quality checked? Were these assemblies conducted de novo? What was used to pool reads and annotate from a genome or de novo? What was used for novel gene and or/transcript isoforms from assemblies? Was a false discovery rate conducted used? P-value or p-adjusted.

Section 2.5- How many samples for each treatment group used for qPCR validation?

Line 164- What version was used? What was used to assess normality and variance of data to make sure assumptions met?

Line 173: Referred to as batfish previously.

Line 178- What membrane?

Line 180- Italicize P. teira.

Line 180- How was malformation quantified? What percent differences between temperature treatments?

Line 182- What were the differences in mortality?

Line 194- What was considered low-quality? The method to determine this should be in the methods section.

Section 3.3- The PCA needs to be described in more detail here. Descriptive percentages and clustering profiles.

Section 3.5 and throughout the manuscript- Gene names, when in reference to expression, are to be lowercased in italicized. There are no results actually presented in this section that are informative- what are these expression relationships, not just similar trends to RNAseq data.

Lines 227-229- These are descriptive and belong in discussion, not results.

Section 3.6- Are there significant differences? Just use up or down regulated.

Line 243- p-value needed if significant. Also, what percent differences or fold changes seen between treatment groups?

Section 3.8- Same comment as above- significant changes, fold differences, lowercase gene names due to expression assessed.

Line 261- This was not made clear in the results section.

Line 315 and 317- Which PPAR?

Line 341- Swallowtail used again- stay consistent.

Figure 1:

Dah not defined; what are the bars denoting in “e”. What is defined as an “abnormality”?

There are not comparisons made in a-d with regard to temperature. This figure is not very informative as is, only “e”, somewhat (although I suggest this be placed in the supplemental section).

Figure 2:

The text is by far too small to read. Suggest including all pathways in a table in the supplemental section and noting the top five GO pathways for clarity. Also, a better representation would be against fold changes, not gene numbers. Gene numbers does not suggest significant changes in pathways, particularly if they are not annotated.

Figure 3:

These are comparisons between qPCR and RNAseq results, not KEGG.

Figure 4:

These are KEGG pathways. Switch figure legends. Also, surprising how GO pathways are based on hundreds of genes and these mostly less than 10. This shows the need to present GO pathways relative to fold change differences.

Table 1: What are the amplicon lengths?

Table 2: What was used to denote significance? Are these normalized to control first?

Table 3: What are the letters indicating? What about variance within treatments? Are these total fish numbers in all replicates?

Only minor English would be needed if accepted.

Author Response

We gratefully thank the editor and all reviewers for their time spend making their constructive remarks and valuable suggestions, which has significantly improved the quality of the manuscript and has enabled us to improve the manuscript. Each suggested revision and comment brought forward by the reviewers was accurately incorporated and considered. Below the comments of the reviewers are responses point by point, and the revisions are indicated. Modifications are marked with color.

Line 74- The “meat is delicious” is a subjective statement and opinion, should be removed.

Response: Thanks for the suggestions. We have removed this section based on your suggestion.

Lines 74-76- This statement should be referenced.

Response: Thanks for the suggestions. We have revised this section based on your suggestions.

Line 76- What is considered to be extreme weather?

Response: Thanks for the suggestions. We have added this section to the description based on your suggestion.

Line 87- A further description of “nutrient fortification” is needed for clarity.

Response: Thanks for the suggestions. Based on your suggestions we have further described the nutritional fortification.

Line 89- How many eggs? How many fertilized? How were they collected? Were the eggs pooled?

Response: Thanks for the suggestions. Based on your suggestions we have described this section in detail in the text.

Line 99- Were these static or flow through systems?

Response: Thanks for the suggestions. The larvae were cultured in still water for the first 7 days, and then cultured in micro-flow water.

Line 105- At what stage hatch?

Response: Thanks for the suggestions. Floating high-quality fertilized eggs (Appearance of myomere) were screened and transport them to Shenzhen by plane.

Line 109- What type of “bait” and how much were they allowed to eat? Was it calculated per body weight?

Response: Thanks for the suggestions. This section we have added in the manuscript.

Lines 109-110- Not clear what is meant by “ensure sufficient food and satiation feeding.”

Response: Thanks for the suggestions. This section we have added in the manuscript.

Line 111- Not clear what was assessed morphologically or what developmental changes consisted of. More detail needed.

Response: Thanks for the suggestions. In this section we mainly observed growth traits, which have been modified in the manuscript.

Line 113- This suggests that n = 1, when in fact n = 3. Were three separate fish pooled together to make 1 sample for RNA seq, so using 9 fish total?

Response: Thanks for the suggestions. After 32 days (all temperature groups entered the juvenile stage), a total of 9 fish were taken from each temperature group, 3 fish were randomly selected from each cage and anesthetized with eugenol, and 3 fish liver samples were pooled as one sample, frozen in liquid nitrogen, and stored in a –80°C refrigerator

Line 114- Freezer What kits were used for library prep? Were these single or paired end? How many bp?

Response: Thanks for the suggestions. For the transcriptome sequencing process, we employed paired-end sequencing with both ends being 150 bp in length.

Line 131- What was considered and acceptable RIN?

Response: Thanks for the suggestions.

Line 132- Sequences need to be made publicly available.

Response: Thanks for the suggestions. All raw reads were submitted to the Sequence Read Archive (SRA) (accession: PRJNA872865).

Section 2.3: How were raw reads quality checked? Were these assemblies conducted de novo? What was used to pool reads and annotate from a genome or de novo? What was used for novel gene and or/transcript isoforms from assemblies? Was a false discovery rate conducted used? P-value or p-adjusted.

Response: Thanks for the suggestions. According to the comments from you, This part is supplemented in 2.4 Bioinformatics Analysis.

Section 2.5- How many samples for each treatment group used for qPCR validation?

Response: Thanks for the suggestions. Select 3 samples for each temperature group and repeat the experiment three times for each sample to ensure accuracy (technical replicates)

Line 164- What version was used? What was used to assess normality and variance of data to make sure assumptions met?

Response: Thanks for the suggestions. We have added to this section.

Line 173: Referred to as batfish previously.

Response: Thanks for the suggestions.

Line 178- What membrane?

Response: Thanks for the suggestions. It is an egg membrane, and we have corrected the errors in the manuscript.

Line 180- Italicize P. teira.

Response: Thanks for the suggestions. We have corrected the errors in the manuscript.

Line 180- How was malformation quantified? What percent differences between temperature treatments?

Response: Thanks for the suggestions. Mainly observe the external shape of the fish body to evaluate whether it is deformed: such as whether the body is deformed, the integrity and shape of the fins, and whether there are abnormalities in the bones. The difference in deformity rate among different temperature groups is shown in Table 3.

Line 182- What were the differences in mortality?

Response: Thanks for the suggestions. The differences in mortality rates among different temperature groups are shown in Table 3

Line 194- What was considered low-quality? The method to determine this should be in the methods section.

Response: Thanks for the suggestions. We have added to this section. In addition, there may be exogenous contaminants introduced into the sequencing sample, such as bacterial DNA, primer residues, etc. To ensure the accuracy and reliability of the analysis, those poor quality and contaminated readings were removed. Based on a preset quality threshold, reads below the threshold are filtered out or trimmed, and only higher-quality reads are retained for subsequent analysis.

Section 3.3- The PCA needs to be described in more detail here. Descriptive percentages and clustering profiles.

Response: Thanks for the suggestions.

Section 3.5 and throughout the manuscript- Gene names, when in reference to expression, are to be lowercased in italicized. There are no results actually presented in this section that are informative- what are these expression relationships, not just similar trends to RNAseq data.

Response: Thanks for the suggestions. This part is a verification of the transcriptome results by qPCR. Through the qPCR results, we can see that it is consistent with the transcriptome results. Further illustrates the accuracy of the transcriptome.

Lines 227-229- These are descriptive and belong in discussion, not results.

Response: Thanks for the suggestions. This part is a verification of the transcriptome results by qPCR. Through the qPCR results, we can see that it is consistent with the transcriptome results. Further illustrates the accuracy of the transcriptome.

Section 3.6- Are there significant differences? Just use up or down regulated.

Response: Thanks for the suggestions. A differential analysis was performed on this gene, and there was a significant difference at p<0.05.

Line 243- p-value needed if significant. Also, what percent differences or fold changes seen between treatment groups?

 Response: Thanks for the suggestions. According to the comments from you, we have already corrected this error.

Section 3.8- Same comment as above- significant changes, fold differences, lowercase gene names due to expression assessed.

 Response: Thanks for the suggestions. According to the comments from you, we have already corrected this error.

Line 261- This was not made clear in the results section.

 Response: Thanks for the suggestions. According to the comments from you, we have already corrected this error.

Line 315 and 317- Which PPAR?

 Response: Thanks for the suggestions. According to the comments from you, we have already corrected this error.

Line 341- Swallowtail used again- stay consistent.

 Response: Thanks for the suggestions. According to the comments from you, we have already corrected this error.

Figure 1:

Dah not defined; what are the bars denoting in “e”. What is defined as an “abnormality”?

There are not comparisons made in a-d with regard to temperature. This figure is not very informative as is, only “e”, somewhat (although I suggest this be placed in the supplemental section).

 Response: Thanks for the suggestions. the bars denoting in "e" stands for error bars. We are a showcase of the different developmental stages and deformities of P. teira.

Figure 2:

The text is by far too small to read. Suggest including all pathways in a table in the supplemental section and noting the top five GO pathways for clarity. Also, a better representation would be against fold changes, not gene numbers. Gene numbers does not suggest significant changes in pathways, particularly if they are not annotated.

Response: Thanks for the suggestions. The original image has been uploaded.

Figure 3:

These are comparisons between qPCR and RNAseq results, not KEGG.

 Response: Thanks for the suggestions. According to the comments from you, we have already corrected this error.

Figure 4:

These are KEGG pathways. Switch figure legends. Also, surprising how GO pathways are based on hundreds of genes and these mostly less than 10. This shows the need to present GO pathways relative to fold change differences.

 Response: Thanks for the suggestions. According to the comments from you, we have already corrected this error.

Table 1: What are the amplicon lengths?

Response: Thanks for the suggestions. According to the comments from you, we added it.

Table 3: What are the letters indicating? What about variance within treatments? Are these total fish numbers in all replicates?

Response: There are significant differences between different letters. Each temperature group was selected 10 fish at a time to determine the developmental stage.

Reviewer 3 Report

The manuscript (Article) by Liu et al presents the results of temperature effects to growth and metabolism of energy and matter of the high economic value fish species Platax teira (from cultivated brood stock) at early development stage. Authors used a wide range of molecular genetic methods - transcriptomics analysis, DEG, and RT-PCR and others. Results of the study showed the optimal temperature for larvae development of this fish species - 24-27оC and temperature limits («stress») of organism functioning on the metabolic level. The research can be useful in studying of the molecular mechanisms of adaptation of the fish organism during early postlarval development at the level of metabolism of matter and energy in response to changes in environmental temperature. I found the manuscript interesting although there are significant comments to structure and clarity of results presentation in the article.

 Clarifying questions and comments are considered:

 General comments

 The Abstract, Introduction, Conclusion need improved focus and clarity.

 Authors use through the article text different terms «cold stress», «thermal stress», «environmental stress», «heat stress» or «energy and material metabolism», «nutrient metabolism», «sugar metabolism»; «survival rate», «deformity rate», «malformation rate». Uniformity and the explanation of their usage should be introduced where possible. The most commonly used phrase «metabolism of matter and energy» instead «energy and material metabolism».

The use of italics for species names should be checked throughout the text (Line 68; Line 180) and the Latin and common fish species names should be given together (Line 63; Line 68) at first mention

Individual figures are hard to read in PDF format (Figure 2, 4, 7)

Not quite clear notation – 24 vs 21оС and others groups in what cases and by what parameters comparison was made.

I am not native speaker, but to my mind, some sentences are unclear in English. Thus, the paper has to be edited by a native speaker or in special language system.

 Specific comments

Abstract:

Line 19-20: I recommend indicating in this proposal that the growth, development and survival of fish, especially in the early stages of development, is influenced by a complex of environmental factors, among which temperature is one of the most important.

Line 21: It would be more correct to use the phrase «molecular mechanisms» in this sentence (extensively studied, the molecular mechanisms are poorly understood).

Line 25: Usually a sentence does not start with numbers and abbreviations. In concluding sentence of the abstract the information from this sentence is repeated.

Line 27-28: The sentence should be rephrased. It will be better to use a verb «involved» in this context instead «enriched».

I recommend use phrases «It was shown; It was found» to emphasizes that the data were obtained in this work.

 It should be indicated in the abstract that the liver of fish was studied.

Introduction:

The introduction does not sufficiently disclose data about the object of study. Please add more data about the features of the life cycle of this fish species, its early development (transfer from the Materials and Methods section - line 172-175). Please add the limits of the temperature of habitat the studied fish in nature.

The aim of study and hypothesis should be taken out in a separate paragraph and be more clearly.

Need to emphasize the significance of the study results for the selection of optimal temperature conditions for breeding Platax teira in aquaculture.

Materials and methods:

There is no data in the rationale for genes’ choice for this study and the functional significance of the enzymes encoded by these genes in metabolic pathways in the organism. This information should be entered in the text or in the table.

Blurred information about the number of samples used.

On what cycle’s genes were raised in the PCR analysis? This information is not provided.

The primers were obtained using transcriptomics analysis. If so, then this should be emphasized in the text of the article.

Results:

Line 245 TCA cycle - give an explanation of the abbreviation.

Discussion and Conclusion:

The reader does not get a single picture of the obtained results, because the discussion dividing into sections. It will better to expand the conclusion and more clearly and concisely describe the results obtained with an emphasis on changes in the expression of genes of the enzymes of various metabolic pathways (lipid metabolism, carbohydrate metabolism) at different developmental stages of fish (larvae, juvenile, adult).

To my mind, some sentences are unclear in English. Thus, the paper has to be edited by a native speaker or in special language system.

Author Response

We gratefully thank the editor and all reviewers for their time spend making their constructive remarks and valuable suggestions, which has significantly improved the quality of the manuscript and has enabled us to improve the manuscript. Each suggested revision and comment brought forward by the reviewers was accurately incorporated and considered. Below the comments of the reviewers are responses point by point, and the revisions are indicated. Modifications are marked with color.

 Clarifying questions and comments are considered:

 General comments

 The Abstract, Introduction, Conclusion need improved focus and clarity.

 Authors use through the article text different terms «cold stress», «thermal stress», «environmental stress», «heat stress» or «energy and material metabolism», «nutrient metabolism», «sugar metabolism»; «survival rate», «deformity rate», «malformation rate». Uniformity and the explanation of their usage should be introduced where possible. The most commonly used phrase «metabolism of matter and energy» instead «energy and material metabolism».

 Response: Thanks for the suggestions. According to the comments from you, we have already corrected this error.

The use of italics for species names should be checked throughout the text (Line 68; Line 180) and the Latin and common fish species names should be given together (Line 63; Line 68) at first mention

 Response: Thanks for the suggestions. According to the comments from you, we have already corrected this error.

Individual figures are hard to read in PDF format (Figure 2, 4, 7)

 Response: Thanks for the suggestions. The original image has been uploaded.

Not quite clear notation – 24 vs 21оС and others groups in what cases and by what parameters comparison was made.

Response: Thanks for the suggestions. According to the comments from you, we have already corrected this error. Sorry for the inconvenience caused by typos.

I am not native speaker, but to my mind, some sentences are unclear in English. Thus, the paper has to be edited by a native speaker or in special language system.

Response: Thanks for the suggestions. Performed language polishing services with Editage.

 Specific comments

Abstract:

Line 19-20: I recommend indicating in this proposal that the growth, development and survival of fish, especially in the early stages of development, is influenced by a complex of environmental factors, among which temperature is one of the most important.

 Response: Thanks for the suggestions. According to the comments from you, we have already corrected this error.

Line 21: It would be more correct to use the phrase «molecular mechanisms» in this sentence (extensively studied, the molecular mechanisms are poorly understood).

 Response: Thanks for the suggestions. According to the comments from you, we have already corrected this error.

Line 25: Usually a sentence does not start with numbers and abbreviations. In concluding sentence of the abstract the information from this sentence is repeated.

 Response: Thanks for the suggestions. According to the comments from you, we have already deleted it.

Line 27-28: The sentence should be rephrased. It will be better to use a verb «involved» in this context instead «enriched».

 Response: Thanks for the suggestions. According to the comments from you, we have already used «involved».

I recommend use phrases «It was shown; It was found» to emphasizes that the data were obtained in this work.

 It should be indicated in the abstract that the liver of fish was studied.

 Response: Thanks for the suggestions. According to the comments from you, we have already corrected this error.

Introduction:

The introduction does not sufficiently disclose data about the object of study. Please add more data about the features of the life cycle of this fish species, its early development (transfer from the Materials and Methods section - line 172-175). Please add the limits of the temperature of habitat the studied fish in nature.

 Response: In the early stage, we have conducted preliminary research on some basics of Platax teira, but it has not been made public, so we will not describe it here.

The aim of study and hypothesis should be taken out in a separate paragraph and be more clearly.

Need to emphasize the significance of the study results for the selection of optimal temperature conditions for breeding Platax teira in aquaculture.

 Response: Thanks for the suggestions. According to the comments from you, we modified this section.

Materials and methods:

There is no data in the rationale for genes’ choice for this study and the functional significance of the enzymes encoded by these genes in metabolic pathways in the organism. This information should be entered in the text or in the table.

 Response: Thanks for the suggestions. Explained in 3.4. GO and KEGG Annotation Analysis of DEGs and discussion.

Blurred information about the number of samples used.

 Response: Thanks for the suggestions. According to the comments from you, we modified this section. After 32 days (all temperature groups entered the juvenile stage), a total of 9 fish were taken from each temperature group, 3 fish were randomly selected from each cage and anesthetized with eugenol, and 3 fish liver samples were pooled as one sample, frozen in liquid nitrogen, and stored in a –80°C refrigerator

On what cycle’s genes were raised in the PCR analysis? This information is not provided.

 Response: Thanks for the suggestions. The PCR conditions were as follows: initial denaturation at 95 °C for 30 s, followed by 40 cycles at 95 °C for 5 s and 60 °C for 30 s.

The primers were obtained using transcriptomics analysis. If so, then this should be emphasized in the text of the article.

 Response: Thanks for the suggestions. According to the comments from you, we modified this section.

Results:

Line 245 TCA cycle - give an explanation of the abbreviation.

 Response: Thanks for the suggestions. According to the comments from you, we have already corrected this error.

Discussion and Conclusion:

The reader does not get a single picture of the obtained results, because the discussion dividing into sections. It will better to expand the conclusion and more clearly and concisely describe the results obtained with an emphasis on changes in the expression of genes of the enzymes of various metabolic pathways (lipid metabolism, carbohydrate metabolism) at different developmental stages of fish (larvae, juvenile, adult).

 Response: Thank you very much for your valuable feedback and review of our research paper. Based on your suggestions, we will make adjustments to the discussion section in order to provide a clearer overall picture of the obtained results. We will reorganize the discussion content to ensure that readers can better understand our findings. Additionally, we will expand the conclusion section to more explicitly and concisely describe the results, with a particular emphasis on changes in the expression of genes involved in metabolic pathways (such as lipid metabolism and carbohydrate metabolism) at different developmental stages of fish.

Reviewer 4 Report

L33: “increasing” instead of “ increase in”

L34: “growth in the fish. Meanwhile, nutrient metabolism” instead of “growth in the fish, nutrient metabolism”.

L90: delete eggs

L 98: you mentioned “five breeding ponds”, however the experimental design contain four groups each on in single pond.

How did you control the low temperature group (21).

(Weinuo) add full details.

“Ten fish were randomly selected from the cages daily” is this logic how many fish did you used for morphological evaluation during the full course of the study. Or the fish retained a live to the experiment again.

For fish sampling, liver of 3 fish from each replicate were pooled, means we have 3 pooled samples per each treatment. Is it enough for Statistical analysis.

Also, pleas be specific when the sample collected (at the end (what is the time course of the experiment), or periodical).

L 145: don’t start the paragraph with abbreviation “GO”.

In subtitle don’t use abbreviation unless you mentioned it as full name in subtitles also not in the text.

Add the Fig. and Table after the first mentioned in the MS. This is reported in authors guide lines of MDPI.

I suggest to include the dimensionality reduction analysis “the principal component” to the results section. Not as Supplementary Fig S1.

L 205: delete “affected”.

L279” Extremes temperature” instead of “Temperature extremes”

In table 3 could you add standard error. Also, could you perform regression analysis to identify the optimum temperature to have sold conclusion.

Some parts in the introduction and discussion need to be rephrased again to avoid high plagiarism.

the language needs moderate revision

Author Response

We gratefully thank the editor and all reviewers for their time spend making their constructive remarks and valuable suggestions, which has significantly improved the quality of the manuscript and has enabled us to improve the manuscript. Each suggested revision and comment brought forward by the reviewers was accurately incorporated and considered. Below the comments of the reviewers are responses point by point, and the revisions are indicated. Modifications are marked with color.

L33: “increasing” instead of “ increase in”

Response: Thanks for the suggestions. According to the comments from you, we have already corrected this error.

L34: “growth in the fish. Meanwhile, nutrient metabolism” instead of “growth in the fish, nutrient metabolism”.

Response: Thanks for the suggestions. According to the comments from you, we have already corrected this error.

L90: delete eggs

Response: Thanks for the suggestions. According to the comments from you, we have already corrected this error.

L 98: you mentioned “five breeding ponds”, however the experimental design contain four groups each on in single pond.

Response: Thanks for the suggestions. According to the comments from you, we have already corrected this error.

How did you control the low temperature group (21).

Response: The temperature of each culture pond was controlled by a thermostat (Weinuo).

(Weinuo) add full details.

Response: Thanks for the suggestions. We added the device details.

“Ten fish were randomly selected from the cages daily” is this logic how many fish did you used for morphological evaluation during the full course of the study. Or the fish retained a live to the experiment again.

Response: Thanks for the suggestions. We have supplemented the morphological observation content more perfectly. “Every day before feeding, 10 fish were selected from each temperature group to observe the changes in morphology and development.”

For fish sampling, liver of 3 fish from each replicate were pooled, means we have 3 pooled samples per each treatment. Is it enough for Statistical analysis.

Also, pleas be specific when the sample collected (at the end (what is the time course of the experiment), or periodical).

Response: Thanks for the suggestions. We supplement the sampling method in detail. “After 32 days (all temperature groups entered the juvenile stage), a total of 9 fish were taken from each temperature group, 3 fish were randomly selected from each cage and anesthetized with eugenol, and 3 fish liver samples were pooled as one sample, frozen in liquid nitrogen, and stored in a –80°C refrigerator”

L 145: don’t start the paragraph with abbreviation “GO”.

Response: Thanks for the suggestions. According to the comments from you, we have already corrected this error.

In subtitle don’t use abbreviation unless you mentioned it as full name in subtitles also not in the text.

Response: Thanks for the suggestions. According to the comments from you, we have already corrected this error.

Add the Fig. and Table after the first mentioned in the MS. This is reported in authors guide lines of MDPI.

Response: Thanks for the suggestions. According to the comments from you, we have already corrected this error.

I suggest to include the dimensionality reduction analysis “the principal component” to the results section. Not as Supplementary Fig S1.

Response: Thanks for the suggestions. According to the comments from you, we put "the principal component" in the manuscript.

L 205: delete “affected”.

Response: Thanks for the suggestions. According to the comments from you, we have already corrected this error.

L279” Extremes temperature” instead of “Temperature extremes”

Response: Thanks for the suggestions. According to the comments from you, we have already corrected this error.

In table 3 could you add standard error. Also, could you perform regression analysis to identify the optimum temperature to have sold conclusion.

Response: Thanks for the suggestions. In this study, a regression analysis was performed to determine the optimal temperature.

Some parts in the introduction and discussion need to be rephrased again to avoid high plagiarism.

Response: Thanks for the suggestions. According to the comments from you, we have already corrected this error.

Reviewer 5 Report

The study by Liu et al analyzed the impact of varying temperatures on P. teira fish's growth, survival, and gene expression using RNA-sequencing. I appreciate the detailed results, methods an discussion the authoer provided. The paper is well structured and has a clear main result that the optimal development at 24-27℃, while extreme temperatures negatively affect physiology and metabolism, informing better aquaculture practices.   I have just a few minor suggestions for impriving its readbiltiy.  
  1. In Method section 2.3, please detail the process of retrieving gene names. Was it via an existing gene annotation or annotations from related model species? Given that these names form the foundation of subsequent analyses, it is important to clarify this process.
  2. In the abstract, I suggest removing the phrase "24-27...development" from lines 25-26, as it is redundant with the last sentence.
  3. Please consider deleting "However" at line 56 for better logical coherence.
  4. On lines 304-305, a minor grammatical correction is needed - please change "These results indicate" to "These results indicate."
  5. About the broader implications: it would be beneficial to discuss briefly how these findings enhance our understanding of temperature effects on fish growth and development. What are the implications for other fish species or for fish farming and conservation efforts? Including a sentence on broader implications in the abstract would greatly improve its impact.

Author Response

We gratefully thank the editor and all reviewers for their time spend making their constructive remarks and valuable suggestions, which has significantly improved the quality of the manuscript and has enabled us to improve the manuscript. Each suggested revision and comment brought forward by the reviewers was accurately incorporated and considered. Below the comments of the reviewers are responses point by point, and the revisions are indicated. Modifications are marked with color.

  1. In Method section 2.3, please detail the process of retrieving gene names. Was it via an existing gene annotation or annotations from related model species? Given that these names form the foundation of subsequent analyses, it is important to clarify this process.

Response: Thanks for the suggestions. We supplement this section in detail in Materials Methods 2.4.

  1. In the abstract, I suggest removing the phrase "24-27...development" from lines 25-26, as it is redundant with the last sentence.

Response: Thanks for the suggestions. According to the comments from you, we delete "24-27...development".

  1. Please consider deleting "However" at line 56 for better logical coherence.

Response: Thanks for the suggestions. According to the comments from you, we have already corrected this error.

  1. On lines 304-305, a minor grammatical correction is needed - please change "These results indicates" to "These results indicate."

Response: Thanks for the suggestions. According to the comments from you, we have already corrected this error.

  1. About the broader implications: it would be beneficial to discuss briefly how these findings enhance our understanding of temperature effects on fish growth and development. What are the implications for other fish species or for fish farming and conservation efforts? Including a sentence on broader implications in the abstract would greatly improve its impact.

Reviewer 6 Report

The study by Ming-Jian Liu et alii examines the effects of breeding temperature on early development and nutrient metabolism in long-finned batfish using transcriptomic techniques. I only have the following minor suggestions for improvement, reflecting the generally high quality of the submitted manuscript. Nevertheless, the language quality needs to be improved throughout as can be seen based on several examples listed below:

-Line 22: “have” instead of “has”

-Line 28: “most of” instead of “most”

-Line 44: This statement is too generalized as regional endothermy has evolved in several large predatory spp., such as the Great White Shark. Please see, e.g., Dickson KA, Graham JB. Evolution and consequences of endothermy in fishes. Physiol Biochem Zool. 2004;77(6):998-1018. doi: 10.1086/423743.

-Line 51:”are crucial periods” instead of “are a crucial periods”

-Line 62 and others throughout the manuscript: species authorities should be indicated upon first mentioning of any species name. Depending on the journal’s preferences, this might be done without indicating the year of the original description (usually the reference is not cited in the list of references then) or with indication of the year of the original description, with the reference to be included in or excluded from the list of references.

-Line 68: add scientific name and species authority for Nile tilapia

-Line 90: delete “eggs”

-Lines 100–101: do you mean previous experiments? And were these also covered by the ethical approval?

-Line 109: “fed” instead of “feed”. Furthermore, “bait” is not necessary and can be deleted.

-Lines 112–113: “cage and anesthetized” instead of “cage, anesthetized”

-Line 159: delete “and”

-Line 173: please explain what you mean with “swallowtail”

-Line 189: “higher than that of downregulated” instead of “higher than the downregulated”

-Line 219: “PPAR” needs to be explained (it is explained in line 223 but should be explained upon first mentioning and the explanation can then be deleted from line 223).

-Discussion, 4.1: are there no other studies examining the effect on temperature of the development of Platax teira or other batfish species? If there are no, this information might be added in order to underline the novelty of the findings but if there are any, they should be mentioned and the results critically compared with those of the present study.

-Lines 274–277: it should be mentioned, which species was investigated by Jin et al. (2019).

-Lines 277–279: the species examined by Lee et al. (2017) should be mentioned.

-Line 256: delete the comma.

-Line 258: The genus name should be written in full when first mentioned in a new manuscript section.

-Line 287: delete the comma.

-Line 335: delete the comma.

-Lines 350–351: as mentioned earlier, it is unclear what is meant with swallowtail.

-Line 407: the genus name should be written in full.

The language quality needs to be improved throughout the manuscript. Several examples can be found in the comments and suggestions listed above.

Author Response

We gratefully thank the editor and all reviewers for their time spend making their constructive remarks and valuable suggestions, which has significantly improved the quality of the manuscript and has enabled us to improve the manuscript. Each suggested revision and comment brought forward by the reviewers was accurately incorporated and considered. Below the comments of the reviewers are responses point by point, and the revisions are indicated. Modifications are marked with color.

-Line 22: “have” instead of “has”

Response: Thanks for the suggestions. According to the comments from you, we have already corrected this error.

-Line 28: “most of” instead of “most”

Response: Thanks for the suggestions. According to the comments from you, we have already corrected this error.

-Line 44: This statement is too generalized as regional endothermy has evolved in several large predatory spp., such as the Great White Shark. Please see, e.g., Dickson KA, Graham JB. Evolution and consequences of endothermy in fishes. Physiol Biochem Zool. 2004;77(6):998-1018. doi: 10.1086/423743.

 Response: Thanks for the suggestions. According to the comments from you, we have modified this content.

-Line 51:”are crucial periods” instead of “are a crucial periods”

Response: Thanks for the suggestions. According to the comments from you, we have already corrected this error.

-Line 62 and others throughout the manuscript: species authorities should be indicated upon first mentioning of any species name. Depending on the journal’s preferences, this might be done without indicating the year of the original description (usually the reference is not cited in the list of references then) or with indication of the year of the original description, with the reference to be included in or excluded from the list of references.

  Response: Thanks for the suggestions. According to the comments from you, we have already corrected this error.

-Line 68: add scientific name and species authority for Nile tilapia

 Response: Thanks for the suggestions. According to the comments from you, we have already corrected this error.

-Line 90: delete “eggs”

 Response: Thanks for the suggestions. According to the comments from you, we have already corrected this error.

-Lines 100–101: do you mean previous experiments? And were these also covered by the ethical approval?

 Response: Thanks for the suggestions. Previous experiments are also included in the scope of ethical approval. According to your suggestion, we have modified this part and put the content approved by the ethics committee on line 112-115

-Line 109: “fed” instead of “feed”. Furthermore, “bait” is not necessary and can be deleted.

 Response: Thanks for the suggestions. According to the comments from you, we have already corrected this error.

-Lines 112–113: “cage and anesthetized” instead of “cage, anesthetized”

 Response: Thanks for the suggestions. According to the comments from you, we have already corrected this error.

-Line 159: delete “and”

 Response: Thanks for the suggestions. According to the comments from you, we have already corrected this error.

-Line 173: please explain what you mean with “swallowtail”

 Response: Thanks for the suggestions. According to the comments from you, we have already corrected this error.

-Line 189: “higher than that of downregulated” instead of “higher than the downregulated”

 Response: Thanks for the suggestions. According to the comments from you, we have already corrected this error.

-Line 219: “PPAR” needs to be explained (it is explained in line 223 but should be explained upon first mentioning and the explanation can then be deleted from line 223).

 Response: Thanks for the suggestions. According to the comments from you, we have already corrected this error.

-Discussion, 4.1: are there no other studies examining the effect on temperature of the development of Platax teira or other batfish species? If there are no, this information might be added in order to underline the novelty of the findings but if there are any, they should be mentioned and the results critically compared with those of the present study.

 Response: Thanks for the suggestions. According to the comments from you, we added "No effect of temperature on the early developmental stages of Platax teira or other batfish species has been found in the existing reports." to the manuscript.

-Lines 274–277: it should be mentioned, which species was investigated by Jin et al. (2019).

  Response: Thanks for the suggestions. Based on your suggestion has been added, the species investigated by Jin et al. (2019).

-Lines 277–279: the species examined by Lee et al. (2017) should be mentioned.

  Response: Thanks for the suggestions. Based on your suggestion has been added, the species investigated by Lee et al. (2017).

-Line 256: delete the comma.

  Response: Thanks for the suggestions. According to the comments from you, we have already corrected this error.

-Line 258: The genus name should be written in full when first mentioned in a new manuscript section.

 Response: Thanks for the suggestions. According to the comments from you, we have already corrected this error.

-Line 287: delete the comma.

  Response: Thanks for the suggestions. According to the comments from you, we have already corrected this error.

-Line 335: delete the comma.

  Response: Thanks for the suggestions. According to the comments from you, we have already corrected this error.

-Lines 350–351: as mentioned earlier, it is unclear what is meant with swallowtail.

  Response: Thanks for the suggestions. According to the comments from you, we have already corrected this error.

-Line 407: the genus name should be written in full.

 Response: Thanks for the suggestions. According to the comments from you, we have already corrected this error.

Round 2

Author Response

Thank you for taking into account previous comments, the manuscript is clearer. However, there is still some typo and the transcriptomic workflow remains elusive.

This comment in the first reviewing round was not fully addressed:

The transcriptomic workflow needs some clarifications and more details, especially concerning the read alignment, did the authors perform a de novo assembly? It is unclear for a reader unfamiliar with non-model organisms lacking a genome.

Same for the comment about the p-value or adjusted p-value for the differential analysis.

Response: Thank you very much for your valuable comments. Based on your suggestion, we will provide further clarification and elaboration on the transcriptome workflow in the Methods section, especially regarding read alignment and whether de novo assembly was performed.

Comments regarding p-values or corrected p-values for differential analyzes are also explicitly stated in the Methods. The modified content is in "2.3. Differential Gene Expression Analysis"

L177: Did the authors could provide data regarding the fact that larvae were most active under high temperatures?

Response: Thanks for the suggestions. This has been supplemented in the manuscript.

The authors speak now about swimming speed, could you provide data for this result? Or is it just an observation? This result is not addressed in the discussion.

Response: Thank you very much for your valuable comments, we have removed this part from the Results in order to improve the logic of the manuscript.

The 3.3 section should be added to the M&M just after the sequencing as it is part of the quality control to perform subsequent data analysis (DEG and enrichment).

Please provide the entire list of significant DEGs, GO and KEGG annotations in supplementary data.

Response: Thanks for the suggestions. The required data for this study, which involves the analysis of a de novo transcriptome, have all been presented.

The table are not in the supplementary data provided by the editor. I understand here that you performed a de novo annotation. Please stressed it in the manuscript.

Response: As per your request, we have added "Complete list of DEG, GO and KEGG annotations" in Supplementary Data. At the same time, we emphasized that the transcriptome result is de novo assembly, and the modifications can be accommodated as follows: "To obtain comprehensive transcriptomic information, de novo assembly was per-formed using the obtained raw sequencing reads. This allowed us to construct a tran-scriptome reference for subsequent analysis.”

Line 109 and 114: Remove the capital letter at the beginning of the brackets

Response: Thanks for your suggestion, we have corrected this error.

Line 115: remove “them”

Response: Thanks for your suggestion, we have corrected this error.

Line 118: the sentence “Select 10,000…” is not clear or at least not in the right tense please rephrase it

Response: Thanks for your suggestion, we have corrected this error.

Line 131: “to ensure sufficient” is repeated

Response: Thanks for your suggestion, we have corrected this error.

Line 176: close the bracket for the GO analysis

Response: Thanks for your suggestion, we have corrected this error.

The x-axis of the 3 figures in Figure 3 is very small. Could the authors please address this issue? If not, maybe reducing the number of terms displayed could be a solution. Idem for Figure 5. For Figure 5 precise if it is a selection of KEGG or all the KEGG pathway enriched.

Response: Thanks for your suggestion, Figure 3 and Figure 5 have been modified according to your requirements, and Figure 5 is the selection of the top 25 KEGG enrichment pathways (the legend has been modified).

Figure 7: So it a heat map of row-normalized log10 (FPKM) values and not DEGs.

Response: Thanks for your suggestion, we have corrected this error. and the modifications can be accommodated as follows: " Fig 7. Heat map illustrating the row-normalized log10(FPKM) values of genes related to glucose metabolism in the liver of P. teira at different temperatures. 

NOTE: Heatmaps represents the expression levels of genes, rather than specifically highlighting the differentially expressed genes (DEGs). The point color shows different Q values.”

Table 4: please add the same note regarding the letters as in table 3

Response: Thanks for your suggestion, we have corrected this error.

Reviewer 2 Report

Not addressed:

What was considered and acceptable RIN?

How were raw reads quality checked? Were these assemblies conducted de novo? What was used to pool reads and annotate from a genome or de novo? What was used for novel gene and or/transcript isoforms from assemblies? Was a false discovery rate conducted used? P-value or p-adjusted.

There is no variance noted in morphological data presented.

Amplicon length not included- bp length not really informative.

What was considered low-quality? The method to determine this should be in the methods section.

            -What parameters were used?

Section 3.6- Are there significant differences? Just use up or down regulated.

            -Upregulation or downregulation?

Figure 2:

The text is by far too small to read. Suggest including all pathways in a table in the supplemental section and noting the top five GO pathways for clarity. Also, a better representation would be against fold changes, not gene numbers. Gene numbers does not suggest significant changes in pathways, particularly if they are not annotated.

            -To emphasize that gene numbers do not suggest significant changes- these need to be annotated.

These are KEGG pathways. Switch figure legends. Also, surprising how GO pathways are based on hundreds of genes and these mostly less than 10. This shows the need to present GO pathways relative to fold change differences.

            -It is not clear how this was corrected as stated?

Table 3: What are the letters indicating? What about variance within treatments? Are these total fish numbers in all replicates?

            -There is no variance presented.

Minor English corrections needed.

Author Response

What was considered and acceptable RIN? 

Response: Thanks for your suggestion, we have corrected this error. We added information about RIN in lines 151 - 153.

How were raw reads quality checked? Were these assemblies conducted de novo? What was used to pool reads and annotate from a genome or de novo? What was used for novel gene and or/transcript isoforms from assemblies? Was a false discovery rate conducted used? P-value or p-adjusted. 

Response: Thanks for your suggestion, we have corrected this error. According to your suggestion, we have made a detailed supplementary explanation in "2.3. Differential Gene Expression Analysis".

There is no variance noted in morphological data presented.

Response: Figure 1 (c) is a normal individual, (d) is a deformed individual

Amplicon length not included- bp length not really informative.

Response: We greatly appreciate the reviewer's feedback. Regarding the length of the primer fragment, we understand and realize that only providing the bp length may indeed not be enough to provide complete information. Can you please indicate what I need to add.

What was considered low-quality? The method to determine this should be in the methods section.

            -What parameters were used?

Response: We have supplemented the instructions in lines 169-171, and the supplementary content is as follows: Use TransRate (http://hibberdlab.com/transrate/) to filter and optimize the sequence of the transcriptome, common errors (including chimeras, structural errors, incomplete assembly, base errors, etc.).

Section 3.6- Are there significant differences? Just use up or down regulated.

            -Upregulation or downregulation?

Response: Significant differences exist and p-values are indicated.

Figure 2:

The text is by far too small to read. Suggest including all pathways in a table in the supplemental section and noting the top five GO pathways for clarity. Also, a better representation would be against fold changes, not gene numbers. Gene numbers does not suggest significant changes in pathways, particularly if they are not annotated.

            -To emphasize that gene numbers do not suggest significant changes- these need to be annotated.

These are KEGG pathways. Switch figure legends. Also, surprising how GO pathways are based on hundreds of genes and these mostly less than 10. This shows the need to present GO pathways relative to fold change differences.

            -It is not clear how this was corrected as stated?

Response: Thanks for your suggestion, we have corrected this error. Revised Figure 3 and Figure 5, As per your request, we have added "Complete list of DEG, GO and KEGG annotations" in Supplementary Data.

Table 3: What are the letters indicating? What about variance within treatments? Are these total fish numbers in all replicates?

            -There is no variance presented.

Response: Thanks for your suggestion, we have corrected this error.

Reviewer 4 Report

the authors improved the manuscript to the standard of publication.

Author Response

Thank you very much for your valuable comments, we have revised the manuscript according to your request.

Round 3

Reviewer 1 Report

Thank you for taking into account for comments. The authors improved the manuscript to the standard of publication.

Reviewer 2 Report

The presentation of results, particularly using gene number to base all results upon, in my opinion, is not correct. There just isn't a clear connection between the biological levels of organization presented, limiting the study impact.